# Structural analysis of an anthrol reductase inspires enantioselective synthesis of enantiopure hydroxycycloketones and β-halohydrins

Xiaodong Hou[1,3], Huibin Xu[1,3], Zhenbo Yuan[1], Zhiwei Deng [1], Kai Fu[1], Yue Gao[1], Changmei Liu[1], Yan Zhang[2] & Yijian Rao [1] ✉

Asymmetric reduction of prochiral ketones, particularly, reductive desymmetrization of 2,2-disubstituted prochiral 1,3-cyclodiketones to produce enantiopure chiral alcohols is challenging. Herein, an anthrol reductase CbAR with the ability to accommodate diverse bulky substrates, like emodin, for asymmetric reduction is identified. We firstly solve crystal structures of CbAR and CbAR-Emodin complex. It reveals that Tyr210 is critical for emodin recognition and binding, as it forms a hydrogen-bond interaction with His162 and π-π stacking interactions with emodin. This ensures the correct orientation for the stereoselectivity. Then, through structure-guided engineering, variant CbAR-H162F can convert various 2,2-disubstituted 1,3-cyclodiketones and *α*-haloacetophenones to optically pure (2S, 3S)-ketols and (R)-β-halohydrins, respectively. More importantly, their stereoselectivity mechanisms are also well explained by the respective crystal structures of CbAR-H162F-substrate complex. Therefore, this study demonstrates that an in-depth understanding of catalytic mechanism is valuable for exploiting the promiscuity of anthrol reductases to prepare diverse enantiopure chiral alcohols.

Chiral alcohols are versatile and valuable synthons for the production of various pharmaceuticals, agrochemicals, and fine chemicals[1–3]. One of the powerful methods for their preparation is enantioselective reduciton of prochiral ketones by chemical or enzymatic asymmetric transfer hydrogenation (ATH)[4–6]. Metal-catalyzed asymmetric transfer hydrogenation (MATH) is the most widely studied way to obtain diverse chiral alcohols, in which expensive and toxic chiral iridium, ruthenium or palladium catalysts are commonly used[7–9]. To circumvent the issues raised by MATH, NAD(P)H-dependent dehydrogenases/reductase-catalyzed ATH of prochiral ketones to produce chiral alcohols has been developed as a green alternative, and the synthesis of many chiral alcohols with high optical selectivity have been successfully realized from aliphatic ketones, alicyclic ketones and aromatic ketones[10–15]. However, enzymes used to synthesize optically pure 2,2-disubstituted-3-hydroxycycloketones from prochiral 2,2-disubstituted 1,3-cyclodiketones through enantioselective desymmetrization are still little reported. This is because 2,2-disubstituted-3-hydroxycycloketones bear two vicinal stereogenic centers and ten isomeric related products could be theoretically afforded without the control of selectivity, thereby representing an challenging problem in synthetic chemistry.

The short-chain dehydrogenases/reductases (SDR), which are widely distributed in both eukaryotes and prokaryotes, are important biocatalysts for the asymmetric reduction of prochiral ketones to afford the corresponding chiral alcohols[10–15]. Among them, anthrol

[1]Key Laboratory of Carbohydrate Chemistry and Biotechnology, Ministry of Education, School of Biotechnology, Jiangnan University, Wuxi 214122, P. R. China. [2]School of Life Sciences and Health Engineering, Jiangnan University, Wuxi 214122, P. R. China. [3]These authors contributed equally: Xiaodong Hou, Huibin Xu ✉e-mail: raoyijian@jiangnan.edu.cn

reductases have gained great interest recently due to their important roles in the biosynthesis of bulky intermediates, such as tricyclic aromatic polyketides, with high stereo- and regioselectivity for the preparation of many natural products (Fig. 1a)[16–19]. For example, emodin hydroquinone, as the natural substrate of most anthrol reductases, can be reduced to (*R*)−3,8,9,10-tetrahydroxy-6-methyl-3,4-dihydroanthracene-1(2*H*)-one (**2a**), a key intermediate for the biosynthesis of many natural products, such as agnestin A[20], ravenelin[21], shamixanthone[22], secalonic acids[23], (-)-flavoskyrin[24], and hemi-cryptosporioptide[25] (Fig. 1a). The reduction of emodin hydroquinone by an anthrol reductase MdpC was firstly achieved in 2012 when emodin (**1a**) was treated with $Na_2S_2O_4$[16], which serves as a reducing agent for the reduction of **1a** to emodin hydroquinone. Then, several other anthrol reductases, including AflM from *Aspergillus parasiticus*[18], 17β-HSDcl from *Curvularia lunata*[26], and ARti and ARti-2 from *Talaromyces islandicus*[19,27], which possess similar catalytic activities as MdpC, were discovered and characterized. However, the enzymatic mechanism of anthrol reductases, especially the structure-function relationship between enzymes and substrates, are still unknown. As they could accommodate bulky substrates for asymmetric reduction, we then aimed to engineer an anthrol reductase not only to address the challenges in efficient reductive desymmetrization of prochiral 2,2-disubstituted 1,3-cyclodiketones in synthetic chemistry, but also to explore other prochiral ketones to understand the limitations of this kind of enzyme.

In this work, an anthrol reductase (CbAR) from fungi *Cercospora* with the ability to efficiently reduce emodin hydroquinone is studied to reveal the catalytic mechanism of anthrol reductases. Crystal structures of CbAR-NADP⁺ and CbAR-NADP⁺-Emodin complex are solved for the first time. It reveals that the loop 209−216 region of

CbAR serves as a "gate" for substrate recognition and binding. Particularly, Tyr210 plays a critical role in precisely regulating emodin recognition, binding and stabilization through the formation of the hydrogen-bond interaction with His162 and π-π stacking interactions with emodin. These interactions ensure the correct orientation of emodin for asymmetric reduction by CbAR. In addition, efficient reductive desymmetrization of bulky 1,3-cyclodiketones to produce optically pure 2,2-disubstituted-3-hydroxycycloketones is successfully realized through structure-guided engineering of CbAR. Various 2,2-disubstituted 1,3-cyclodiketones can be efficiently converted by CbAR-H162F into the corresponding (2*S*, 3*S*)-ketols with excellent enantioselectivity. Furthermore, variant CbAR-H162F can also smoothly catalyze various α-haloacetophenones to afford optically pure β-halohydrins, important synthetic building blocks for various pharmaceuticals. More importantly, their stereoselectivity mechanisms are well explained by their crystal structures of CbAR-H162F-substrate complex.

## Results

### Identification and characterization of an anthrol reductase CbAR with a broad substrate scope

Based on sequence homology screening, CbAR was identified as a putative anthrol reductase from *Cercospora* sp. JNU001, which is supposed to be involved in the biosynthesis of phytotoxin beticolin 1 based on the recent study[28]. It belongs to the SDR superfamily and shares the highest sequence identity of 86.6% with 17β-HSDcl from *Curvularia lunata* (Fig. 1b and Supplementary Table 1). In addition, CbAR has high sequence homology with several typical anthrol reductases (MdpC, AflM, ARti and ARti-2) (Fig. 1b and Supplementary Table 1), which could catalyze **1a** to afford **2a** in the presence of

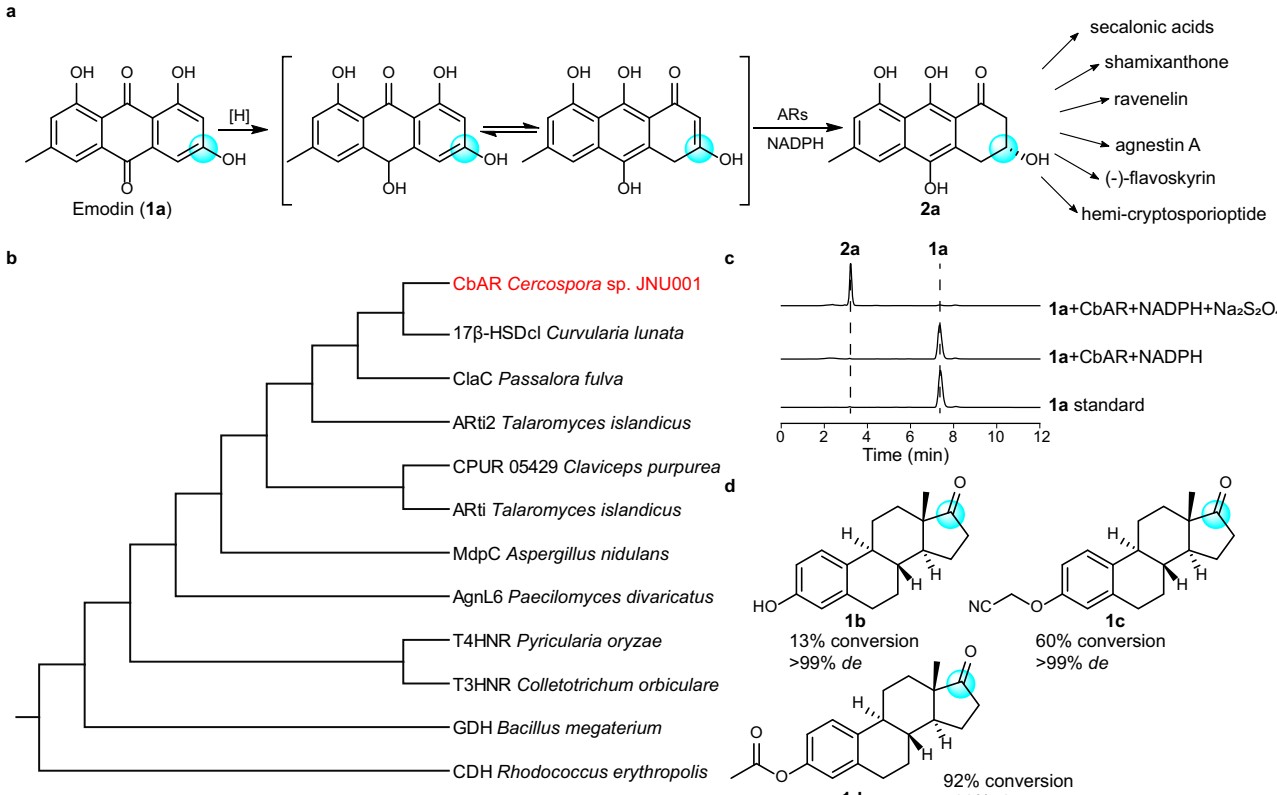

**Fig. 1 | Identification of anthrol reductase CbAR with the ability to catalyze the reduction of emodin and estrones. a** The reduction of emodin (**1a**) catalyzed by anthrol reductases (ARs) in the presence of NADPH and $Na_2S_2O_4$. The reduced product **2a** acts as an important intermediate for the biosynthesis of many natural products. **b** Phylogenetic tree analysis of CbAR. **c** HPLC profiles of CbAR catalyzed reduction of emodin in the presence of NADPH and $Na_2S_2O_4$. **d** Estrone and its analogues could be reduced by CbAR. The reactive positions are highlighted.

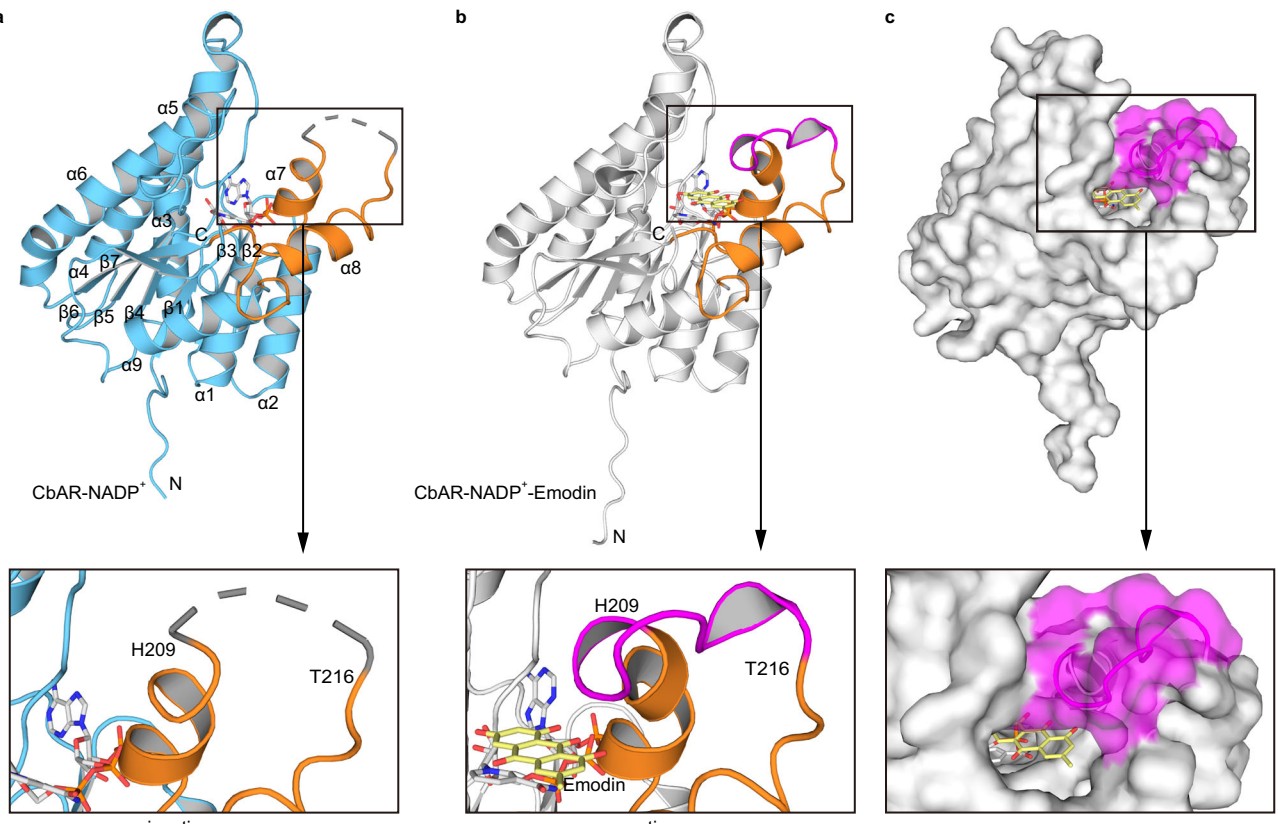

**Fig. 2 | Overall crystal structure of CbAR in complex with NADP⁺ or NADP⁺-Emodin. a** Crystal structure of CbAR-NADP⁺ complex and close-up view of the region from residue 209 to 216 (PDB: 7YB1). The standard Rossmann-fold motif responsible for NADPH binding and the C-terminal domain for substrate-binding are indicated by blue and orange, respectively. The region between residues 209–216 is displayed as a dashed line. **b** Crystal structure of CbAR-NADP⁺-Emodin complex and close-up view of the region from residue 209 to 216 (PDB: 7YB2). The standard Rossmann-fold motif responsible for NADPH binding and the C-terminal domain for substrate-binding are indicated by grey and orange, respectively. The region from residue 209 to 216 is shown in violet. **c** Surface representation profile of CbAR-NADP⁺-Emodin complex and close-up view of the surface of the region between residues 209–216 colored in violet.

$Na_2S_2O_4$ (Fig. 1a, b)[18,19,27], indicating that **1a** is a potential substrate for CbAR. To prove this hypothesis, in the presence of $Na_2S_2O_4$, **1a** was incubated with purified CbAR, which was eluted as a dimer in solution (Supplementary Fig. 1a–c), and nicotinamide adenine dinucleotide phosphate (NADPH) (regenerated through the glucose/glucose dehydrogenase (GDH) system) in 50 mM potassium phosphate buffer under argon atmosphere for 1 h (Fig. 1c and Supplementary Fig. 1). It showed that **2a** was obtained with 99% conversion and >99% *ee* under optimal conditions (pH 8.0, 25 °C) (Fig. 1c and Supplementary Fig. 1d, e). Next, enzyme kinetic parameters of CbAR were analyzed. According to Michaelis-Menten equation (Supplementary Fig. 1f), kinetic parameters of CbAR were calculated, showing that its $K_m$ and $k_{cat}$ values were 0.20 mM and 102.51 min⁻¹ (Supplementary Fig. 1f), respectively. In addition, the melting temperature for CbAR was determined to be 40.3 °C by circular dichroism (Supplementary Fig. 1g, h). These results demonstrate that CbAR is an anthrol reductase with the ability to catalyze the reduction of emodin with high regioselectivity and stereoselectivity.

According to phylogenetic tree analysis (Fig. 1b), CbAR shares a high sequence identity with 17β-HSDcl, which is known to catalyze the reduction of estrone (**1b**)[29]. We then determined whether estrone could be reduced by CbAR. As shown in Figs. 1d, **1b** and its analogues (**1c**, **1d**) could be correspondingly transformed to 17β-estradiols with 99% *de*. In fact, they could be converted from related precursors 2,2-disubstituted-3-hydroxycycloketones[30,31]. Together with the above results, it implies that CbAR has the potential capacity to accommodate diverse bulk prochiral ketones, probably including prochiral 2,2-disubstituted 1,3-cyclodiketones to prepare optically pure chiral alcohols.

## Overall crystal structure of CbAR

Although the reduction of emodin by an anthrol reductase was realized in 2012[16], its catalytic mechanism had not been investigated. To understand the catalytic mechanism of CbAR, crystal structures of CbAR-NADP⁺ and CbAR-NADP⁺-Emodin were then solved and refined to 3.30 Å and 1.85 Å resolutions (Supplementary Table 2), respectively. It shows that the crystallographic asymmetric unit contains four molecules (Supplementary Fig. 2), but CbAR was eluted as a dimer in solution (Supplementary Fig. 1b, c). Similar to other SDRs, the overall structure of CbAR monomer contains nine α-helices and two sets of β-sheets (β1-β7) (Fig. 2a, b), in which continuous β-sheets and seven α-helices constitute the standard Rossmann-fold motif responsible for NADPH binding and the C-terminal domain constitutes a substrate binding pocket (Fig. 2a)[32,33]. Based on structure alignment, although the root mean square difference (RMSD) of superposition for the Cα atoms of crystal structures of CbAR-NADP⁺ and CbAR-NADP⁺-Emodin is 0.25Å, the regions from residue 209 to 216 are clearly different (Fig. 2a, b). Upon the binding of emodin (Supplementary Fig. 3a, b), this region of CbAR-NADP⁺-Emodin ternary complex is visible (Supplementary Fig. 4a), which is absent in the CbAR-NADP⁺ complex due to low electron density, indicating that this region plays a critical role in emodin recognition and binding. Surface representation profile of CbAR-NADP⁺-Emodin complex further

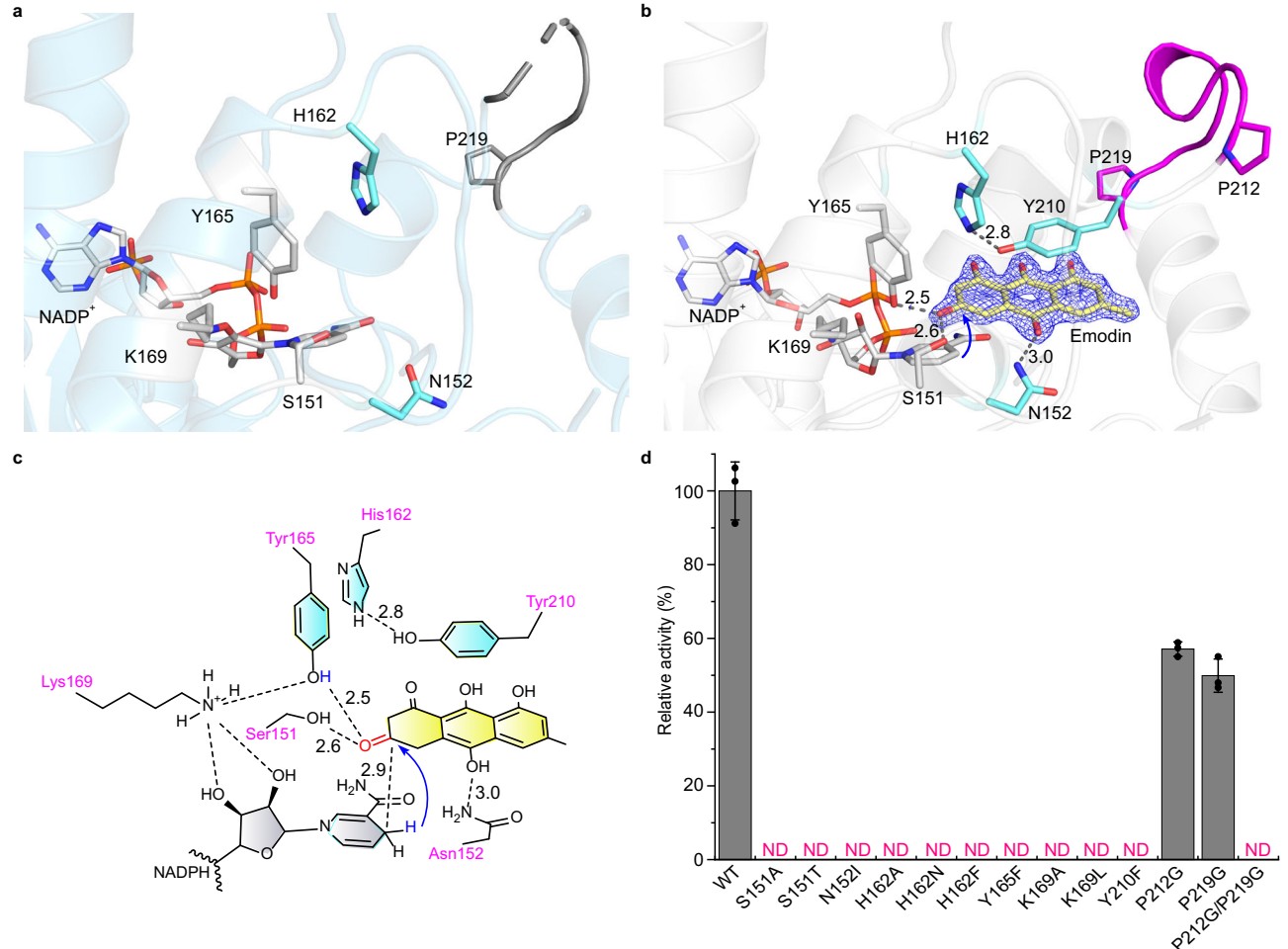

**Fig. 3 | Molecular insights and characterization of emodin recognition and binding of CbAR. a** Substrate binding pocket of CbAR without emodin. NADP⁺, the catalytic triad (Ser151, Tyr165 and Lys169), the flexible loop (between Pro212 and Pro219), and other key residues (Asn152 and H162) are indicated. **b** Substrate binding pocket of CbAR with emodin. NADP⁺, the catalytic triad (Ser151, Tyr165 and Lys169), the flexible loop (between P212 and P219), residues involved in binding of emodin (N152, H162 and Y210) are highlighted in different colors. The 2Fo – Fc electron density maps of emodin was contoured at 1.0 σ in blue color. The proposed proton shuttling mechanism of CbAR towards emodin is indicated by the blue arrow. **c** A detailed interaction network between CbAR and emodin. The proposed proton shuttling mechanism of CbAR towards emodin is indicated by the blue arrow. **d** The relative activity of wild-type (WT) CbAR and its variants towards emodin (expressed as the average of *n* = 3 independent experiments). ND, not detected. Error bars indicate ±sd.

demonstrates the importance of this region, which may serve as a "gate" for emodin recognition and binding (Fig. 2c).

**Molecular insights of emodin recognition and binding of CbAR**
To better understand the importance of the "gate" (the region from residue 209 to 216) for emodin recognition and binding, a detailed interaction between emodin and the surrounding amino acid residues of CbAR was further analyzed. In both of crystal structures of CbAR-NADP⁺ and CbAR-NADP⁺-Emodin, the catalytic triad (Ser151, Tyr165 and Lys169) is well observed (Fig. 3a, b). Ser151 plays a role in stabilizing the substrate. Tyr165 serves as a general base for proton transfer. Lys169 can not only form the hydrogen-bond interaction with NADPH but also reduce the pKa of Tyr-OH (Supplementary Fig. 5)[34]. With the binding of emodin, which is well fitted into the enzyme active site pocket (Fig. 3b and Supplementary Fig. 3), Tyr210 localized at the region from 209 to 216 not only forms the strong hydrogen-bond interaction with His162 with the distance of 2.8Å, but also π-π stacking interactions with emodin (Fig. 3b), suggesting that it precisely regulates emodin recognition, binding and stabilization. Asn152 is also involved in the binding of emodin through the hydrogen bond interaction (Fig. 3b, c). Furthermore, the reactive carbonyl oxygen of emodin, which is supposed to be hydroxyl group during the catalytic reaction (Fig. 1a), is

hydrogen bonded with the side chains of catalytic active residues Ser151 and Tyr165 with distances of 2.6 Å and 2.5 Å, respectively. The above interactions ensure the correct orientation of emodin for asymmetric reduction by CbAR (Fig. 3b, c), which will determine the stereo- and regioselectivity of the reaction due to the direction of nucleophilic attack of the hydrogen atoms at the C4 position on the pyridine ring of NADPH to the carbonyl oxygen of emodin (Fig. 3b, c)[35]. Thus, we conclude that residues Asn152, His162 and Tyr210 play critical roles in recognition, binding and stabilization of emodin.

To further validate the importance of these key residues, site-directed mutagenesis was carried out. As expected, the catalytic activity of CbAR for emodin was inactivated when residues of the catalytic triad (Ser151, Tyr165 and Lys169) were mutated (Fig. 3d). Meanwhile, the reduction activity of CbAR was completely abolished when the hydrogen-bond interaction between His162 and Tyr210 was broken through mutation, that is, His162 was mutated to alanine, asparagine or phenylalanine and Tyr210 was replaced by phenylalanine (Fig. 3d). The catalytic activity of CbAR was also lost when Asn152 was mutated to isoleucine. The results further support the above conclusion that Asn152, His162, and Tyr210 play critical roles in emodin recognition and binding. As for the "gate" region (from residue 209 to 216) (Fig. 2b), besides Tyr210, Ile211 is also critical for the catalytic

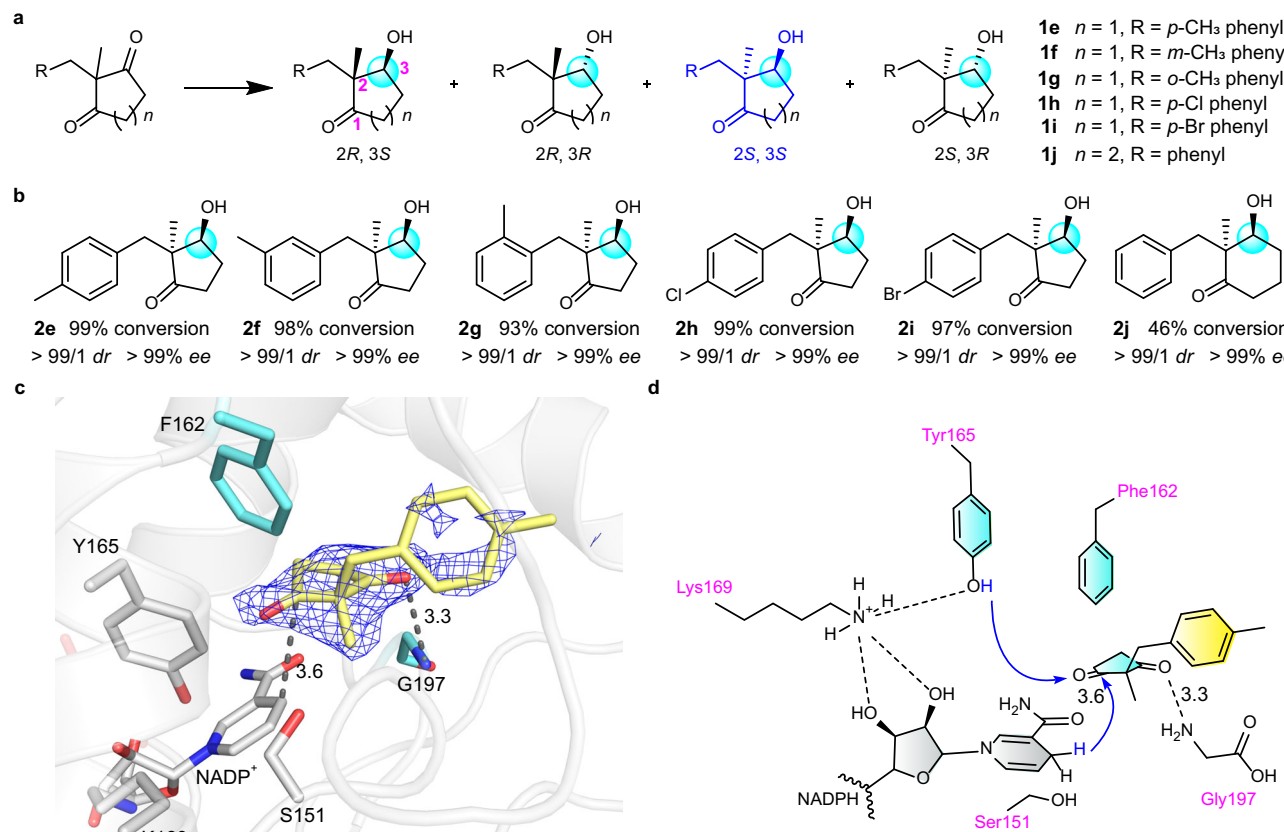

**Fig. 4 | Molecular insights of reductive desymmetrization of 2,2-disubstituted prochiral 1,3-cyclodiketones for the preparation of (2S, 3S)−2,2-disubstituted-3-hydroxycycloketones by CbAR-H162F. a** Four possible stereoisomers are generated by reduction desymmetrization of 2,2-disubstituted-1,3-cyclodiketones. **1e−1j** represent 6 prochiral 1,3-cyclodiketones used in this study. The key carbon atom involved in the asymmetric reduction of 2,2-disubstituted prochiral 1,3-cyclodiketones was highlighted with the light blue circle. **b** Reduced compounds (**2e−2j**) obtained by the reduction of substrates (**1e−1j**) by CbAR-H162F. The key

carbon atom involved in the asymmetric reduction of 2,2-disubstituted prochiral 1,3-cyclodiketones was highlighted with the light blue circle. **c** Substrate binding pocket of CbAR-H162F for substrate **1e**. The 2Fo − Fc electron density map of substrate **1e** was contoured at 1.0 σ in blue color (PDB: 8HFJ). **d** Proposed proton shuttling mechanism and substrate binding mode of CbAR-H162F for the reduction of substrate **1e**. The proposed proton shuttling mechanism of CbAR towards substrate **1e** is indicated by the blue arrow.

activity of CbAR (Supplementary Fig. 6). Interestingly, the catalytic activity of CbAR was impaired when Pro212 or Pro219 was mutated to glycine, and P212A/P219A variant even completely inhibit the catalytic activity (Fig. 3d). The reason could be that loosening the rigidity of the region from residue 209 to 216 would increase the energy for the conformation change of Tyr210 during emodin recognition. These results suggest that Pro212 and Pro219 also play important roles in substrate recognition of emodin (Fig. 3b).

### Structure-guided engineering of CbAR for the preparation of (2S, 3S)−2,2-disubstituted-3-hydroxycycloketones through reductive desymmetrization

Owing to two chiral centers of 2,2-disubstituted-3-hydroxycycloketones, four stereoisomers could be possibly synthesized through reductive desymmetrization of 2,2-disubstituted 1,3-cyclodiketones (Fig. 4a)[12,15]. Thus, it is a great challenge to prepare optically pure 2,2-disubstituted-3-hydroxycycloketones in synthetic chemistry. Since CbAR can accommodate different bulky substrates (Fig. 1d), we next attempted to investigate whether CbAR could convert 2,2-disubstituted 1,3-cyclodiketones into optically pure 2,2-disubstituted-3-hydroxycycloketones through reductive desymmetrization. We then used CbAR as the biocatalyst to conduct the reduction of 2-methyl-2-(4-methylbenzyl)cyclopentane-1,3-dione (**1e**) with co-factor regeneration by a glucose/GDH system. The (2S, 3S)-stereoisomer in >99% ee was afforded (Supplementary Table 3 and Supplementary Fig. 7), showing the excellent stereoselectivity (Fig. 4a, b). Owing to the

importance of (2S, 3S)−2,2-disubstituted-3-hydroxyketones, which could be used for the preparation of many natural products, such as cortistatin A, clavulactone, madindoline A, digitoxigenin and hygrophorone (Supplementary Fig. 8)[36–41], enzyme kinetic parameters of CbAR towards **1e** were then analyzed. It showed that its $K_m$ and $k_{cat}$ values were $5.13 \pm 0.33$ mM and $0.58 \pm 0.01$ min⁻¹ (Supplementary Table 4), respectively, suggesting that the catalytic activity of CbAR towards **1e** is much lower than that towards to **1a** (Supplementary Fig. 1f, h), which could be explained by the precise recognition or orientation of **1a** by Tyr210 (Fig. 3b, c), with the formation of the hydrogen-bond interaction with His162.

Next, Tyr210 and His162 were mutated to break the precise control for emodin recognition (Fig. 3b, c), and then investigated whether it would modulate the catalytic activity of CbAR towards **1e**. Indeed, CbAR-H162F, CbAR-Y210F and CbAR-Y210A variants could enhance the catalytic activity towards **1e**, with 44.7, 4.6 and 3.2-fold increase (Supplementary Table 4), respectively. More importantly, the stereoselectivity of product was not changed and the sole (2S, 3S)-stereoisomer (>99% ee) was still obtained (Supplementary Table 4). Then, CbAR-H162F was selected to explore the substrate scope. Various **1e** analogues with the *meta*- (**1f**) and *ortho*-methyl (**1g**), *para*-chloro (**1h**) and *para*-bromo groups (**1i**) on the benzene ring were synthesized for the substrate scope investigation (Fig. 4a, b). All of them could be efficiently converted into the corresponding (2S, 3S)-ketols (**2e−2i**) with higher than 93% conversion and 99% ee by CbAR-H162F (Fig. 4b and Supplementary Table 3), whose conversions were better than that

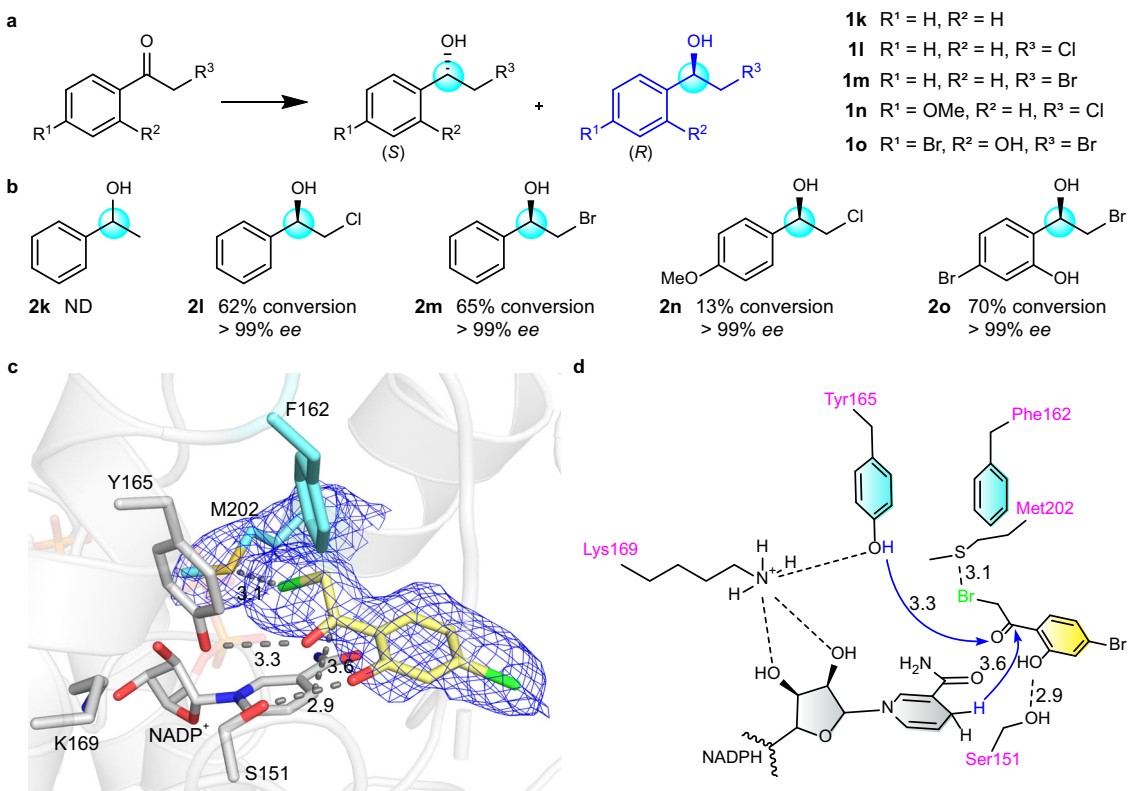

**Fig. 5 | Molecular insights of asymmetric reduction of α-haloacetophenones for the preparation of β-halohydrins by CbAR-H162F. a** Two possible stereoisomers are generated by reduction of α-haloacetophenones. **1k–1o** represents 5 α-haloacetophenones used in this study. The key carbon atom involved in the asymmetric reduction of α-haloacetophenones was highlighted with the light blue circle. **b** Reduced compounds (**2k–2o**) obtained by the reduction of substrates (**1k–1o**) by CbAR-H162F. ND, not detected. The key carbon atom involved in the asymmetric reduction of substrates (**1k–1o**) was highlighted with the light blue circle. **c** Substrate binding pocket of CbAR-H162F for substrate **1o** (PDB: 8HFK). Atoms S and Br are colored in yellow and green, respectively. The 2Fo – Fc electron density map of substrate **1o** and Met202 was contoured at 1.0 σ in blue color. **d** Proposed proton shuttling mechanism and substrate binding mode of CbAR-H162F for the reduction of substrate **1o**. The proposed proton shuttling mechanism of CbAR towards substrate **1o** is indicated by the blue arrow.

of wild-type CbAR (Supplementary Table 3). Furthermore, optically pure (2S, 3S)−2-benzyl-3-hydroxy-2-methylcyclohexan-1-one (**2j**) could also be obtained from the related six membered 1,3-diketone (**1j**) catalyzed by CbAR-H162F (Fig. 4b).

To reveal the exclusive stereoselectivity for 2,2-disubstituted 1,3-cyclodiketones by CbAR-H162F, crystal structure of CbAR-H162F-NADP⁺-**1e** complex was successfully obtained and refined to 2.75Å resolution (Supplementary Table 2). Its overall structure is very similar with CbAR-emodin complex. The "gate" region (from residue 209 to 216) is visible (Supplementary Fig. 4b), but the conformation is different (Supplementary Fig. 9a, b). Importantly, although the electron density map of 4-methylbenzyl group of **1e** was unclear, probably due to the considerable mobility of the flexible benzyl groups, the 2-methyl-cyclopentane-1,3-dione part of **1e** is clearly observed and well localized to the active site of CbAR-H162F (Fig. 4c, d and Supplementary Fig. 3c, d). Notably, we observed that only two of the possible four binding sites in the crystallographic asymmetric unit are occupied by **1e**. One carbonyl group of **1e** forms the hydrogen-bond interaction with the amino group of Gly197 (Fig. 4c, d). In addition, it shows that the 2-methyl group of **1e** orients towards co-factor NADP⁺ (Fig. 4c). In all, these observations well demonstrate the exclusive stereoselectivity for 2,2-disubstituted 1,3-cyclodiketones catalyzed by CbAR-H162F to deliver corresponding (2S, 3S)-ketols.

**The construction of optically pure β-halohydrins through asymmetric reduction of α-haloacetophenones by CbAR-H162F**

Last, we tried to explore other prochiral ketones to understand the limitations of CbAR and CbAR-H162F. Acetophenone (**1k**) and

2-chlorocetophenone (**1l**) were selected to investigate whether they could be converted to related enantiopure chiral alcohols (Fig. 5a), which have been widely applied to synthesize natural products, fine chemicals and pharmaceuticals[42–44]. It shows that no new products (**2k**, **2l**) were detected for both substrates catalyzed by wild-type CbAR (Supplementary Table 5). However, 2-chloroacetophenone (**1l**) could be efficiently reduced to **2l** with excellent enantioselectivity (>99% *ee*) by CbAR-H162F (Fig. 5b and Supplementary Table 5). Furthermore, it was found that 2-bromoacetophenone (**1m**) and haloacetophenones containing different substituents on the benzene ring (4-methoxy (**1n**), 2-hydroxyl-4-bromo (**1o**)) were also compatible in this reaction. They delivered the corresponding products **2m**, **2n** and **2o** smoothly with high enantioselectivity (>99% *ee*) (Fig. 5b and Supplementary Table 5). These results suggest that the engineered CbAR-H162F is a versatile reductase for a broad range of prochiral ketones.

To understand why 2-chlorocetophenone (**1l**), but not acetophenone (**1k**), could be efficiently biotransformed into β-halohydrins with excellent enantioselectivity, the crystal structure of CbAR-H162F/α-haloacetophenone complex was also investigated. Fortunately, the crystal structure of CbAR-H162F-NADP⁺-**1o** complex was successfully solved and refined to 2.90Å resolution (Fig. 5c, d, Supplementary Fig. 2 and Supplementary Table 2). The "gate" region is also visible (Supplementary Fig. 4c, 9c), but the conformation is different from CbAR-emodin complex or CbAR-H162F-NADP⁺-**1e** complex (Supplementary Fig. 9a). The substrate **1o** well orients towards the active site of CbAR-H162F through the formation of hydrogen-bond interactions with Ser151 and Tyr165 (Fig. 5c and Supplementary Fig. 3e, f), in which the interaction between Tyr165−OH and **1o** will facilitate hydride transfer

for the reduction. Moreover, their interaction is further stabilized by the sulfur-halogen bond interaction between Met202 and the bromo group of **1o** with the distance of 3.1Å (Fig. 5c). Actually, this sulfur-halogen bond has been observed in many crystal structures and used for drug discovery[45–47]. This result well explains why α-haloacetophenones, but not acetophenone, were able to be converted to β-halohydrins with excellent enantioselectivity.

## Discussion

In this study, an anthrol reductase (CbAR) from *Cercospora* sp. JNU001 with the ability for asymmetric reduction of various bulky substrates was identified and characterized. In order to reveal the catalytic mechanism of CbAR, crystal structures of CbAR-NADP⁺ and CbAR-NADP⁺-Emodin complex were solved for the first time. It reveals that the loop 209–216 region of CbAR acted as a "gate" for emodin recognition and binding through the force of the strong hydrogen-bond interaction between His162 and Tyr210, and well explained the stereoselectivity control mechanism. More importantly, through breaking down the interaction between His162 and Tyr210, variant CbAR-H162F could not only improved the catalytic activity of CbAR, with 44.7-fold increase, but also convert various 2,2-disubstituted 1,3-cyclodiketones into the corresponding (2*S*, 3*S*)-ketols with excellent enantioselectivity by reductive desymmetrization, which is an challenging problem in synthetic chemistry. Furthermore, CbAR-H162F could also smoothly reduce various α-haloacetophenone to afford optically pure β-halohydrins with excellent enantioselectivity. More importantly, their mechanisms for excellent stereoselectivity were well revealed by the respective crystal structures of CbAR-H162F-NADP⁺ substrate complexes. Therefore, this study suggests the great potential for engineering anthrol reductases to exploit their promiscuity, and then to prepare complex compounds with multiple chiral centers with high enzyme activity and stereoselectivity.

## Methods

### Materials

Anthrol reductase (CbAR) is mined from *Cercospora* sp. JNU001 which was isolated in our laboratory. Primers used for gene amplification in this research were synthesized by Yi Xin Life Technologies (Wuxi, China). PrimeSTAR Max DNA Polymerase (2×) was obtained from Takara Ltd (Shanghai, China). The M-Quick cloning kit, which was used to ligate DNA fragments, was bought from Yi Xin Life Technologies (Wuxi, China). All other reagents and solvents used in this study were purchased from Sigma-Aldrich (Shanghai, China), China National Pharmaceutical Group Corp (Shanghai, China), Energy Chemical (Shanghai, China) and Bidepharm (Shanghai, China).

### Cloning, expression, and purification of recombinant CbAR and its variants

The *CbAR* gene without introns from *Cercospora* sp. JNU001 was amplified using the primers 5′- GGAGATATACATATGGCTATGTC GCCACCAACACAAGAC-3′ and 5′- GTGGTGGTGCTCGAGTGCCGCAGC ACCACCATCTACAG-3′. Then the fragment was purified and ligated with the pET21b (+) expression vector, resulting in a His₆ tag at the C-terminal, and then used for protein expression. Generally, overnight cultures are inoculated to 1L 2×YT medium containing 100 μg/mL ampicillin. Protein expression was induced with 0.1 mM β-D-1-thiogalactopyranoside (IPTG) at 18 °C for 12 h at $OD_{600}$ = 0.6–0.8.

Next, the cells were collected, resuspended with lysis buffer (25 mM Tris-HCl pH 8.0, 300 mM NaCl and 5% (w/v) glycerol), and then lysed by a high pressure homogenization system (Union-Biotech Co., Ltd., Shanghai, China). The cell lysates were centrifuged at 40,000 × *g* for 30 min at 4 °C. The supernatant was recovered and loaded onto the NI-NTA agarose column which was pre-equilibrated with buffer A (25 mM Tris-HCl pH 7.4, 300 mM NaCl and 20 mM imidazole). After extensive wash with buffer A, protein samples were eluted by buffer B (25 mM Tris-HCl pH 7.4, 300 mM NaCl and 250 mM imidazole). After concentration and filtration, the protein sample was applied to a Hiload 16/600 superdex 200 pg column (GE Healthcare), which was pre-equilibrated with running buffer (25 mM Tris-HCl pH 8.0 and 150 mM NaCl). The purified CbAR or its variants was stocked at −80 °C for further analysis. The primers used for the constructed mutant vectors are listed in Supplementary Table 6.

### Enzymatic activity assay

The catalytic activities of CbAR and its variants towards emodin were determined as follows. All buffers were degassed under reduced pressure for 20 min in nitrogen atmosphere. A 100 μL mixture consisting of D-glucose (10 mM), NADP⁺ (1 mM), $Na_2S_2O_4$ (40 mM), substrate (2 mM), GDH (6.8 μM), and CbAR (1.7 μM) were mixed in 500 μL EP tube and then transferred to a 25 mL Schlenk tube under nitrogen atmosphere. The reaction tube was stirred at 25 °C for 20 min under nitrogen atmosphere. The reaction solution was extracted with EtOAc, and the solvent was evaporated under vacuum. The sample was dissolved in acetonitrile and analyzed by HPLC. The HPLC chromatographic conditions were shown in Supplementary Table 7.

For catalytic activity assay of CbAR and its variants towards 1,3-cyclodiketones, estrone and their derivatives, the reaction system (100 μL) containing D-glucose (10 mM), NADP⁺ (1 mM), GDH (6.8 μM), CbAR (10 μM) and substrates (2 mM) was performed at 25 °C. The reaction times for estrone and 1,3-cyclodiketones were 12 h and 7 h, respectively. Then the reaction was quenched by acetonitrile and analyzed by HPLC. The HPLC chromatographic conditions were shown in Supplementary Table 7.

As for the catalytic activity assays of CbAR and its variants towards α-haloacetophenones, the reaction system (100 μL) containing NADPH (5 mM), CbAR (35 μM) and substrates (2 mM) was performed at 25 °C for 12 h. Then, the reaction was quenched by acetonitrile and analyzed by HPLC. The HPLC chromatographic conditions were shown in Supplementary Table 7.

### Melting temperature (Tm), optimum pH and temperature of CbAR

To determine the melting temperature (Tm) of CbAR, the ellipticity of CbAR (0.2 mg/mL in 25 mM Tris-HCl, pH 8.0) was observed as a function of temperature (20–80 °C) at 220 nm. Then, the obtained data were fitted to the Boltzmann equation, and the melting temperature (Tm) was obtained.

The optimum pH and temperature: The conversion of emodin was used for screening optimum pH and optimum temperature for CbAR-catalyzed reduction of emodin hydroquinone. The experimental procedure and sample analysis method are similar to the catalytic activity assay. Potassium phosphate buffer (pH 6.0–8.0, 50 mM) and Tris–HCl buffer (pH 8.0–9.0, 50 mM) were used to determine the optimum pH. The reaction temperature was set from 15 °C to 50 °C with an interval of 5 °C to obtain the optimum temperature.

### Kinetic assay

Kinetic parameters ($V_{max}$, $K_m$ and $k_{cat}$) were determined by measuring the initial rates at various substrate concentrations at 25 °C in 50 mM potassium phosphate buffer, pH 8.0. The experimental procedure and the sample analysis method are similar to the catalytic activity assay. The Michaelis-Menten equation was utilized to obtain the kinetic parameters.

### Crystallization and x-ray structure analysis

To obtain crystals of CbAR-NADP⁺ complex, CbAR was concentrated to 5–6 mg/mL in running buffer containing 2 mM NADP⁺ and then screened using commercial crystallization kits, such as PEGRx, Index HT, PEG-Ion, Classics suite and MbClass Suite. Equal volumes of protein sample and reservoir solution were mixed and added to each drop

well of a 96 well plate at 18 °C. After screening and further optimization, diffraction-quality crystals were obtained with the condition: 0.1M HEPES pH 7.5, 15% isopropanol and 20% PEG4000 at 18 °C. To obtain crystals of CbAR-NADP⁺-Emodin complex. The protein sample was concentrated to 5–6 mg/mL and incubated with 2 mM NADP⁺ and 2 mM emodin overnight. Then protein samples were used for crystallization. After 3–5 days, red crystals of CbAR-NADP⁺-Emodin complexes appeared in same condition mentioned above. To obtain crystals of CbAR-H162F-**1e** complex and CbAR-H162F-**1o** complex, the protein sample was concentrated to 5–6 mg/mL and incubated with 2 mM NADP⁺ and 10 mM substrates **1e**/**1o** overnight. Protein samples were then used for crystallization. After 2 to 3 days, crystals of CbAR-H162F-NADP⁺-**1e** and CbAR-H162F-NADP⁺-**1o** complexes appeared in 0.1M HEPES pH 7.5, 15% isopropanol and 19% PEG4000 at 18 °C.

For data collection, crystals were cryoprotected by transient soaking in appropriate reservoir solutions containing 20–25% glycerol and rapidly cooled in liquid nitrogen. Data collection was performed at the BL19U1 beamline at SSRF (Shanghai, China) or at an in-house Bruker D8 Venture. The HKL-2000 program suite (data set from SSRF)[48] or PROTEUM3 (data set from Bruker D8 Venture) was utilized to process the data set.

The phases of CbAR and CbAR-H162F variant were solved using Molrep-auto MR in the CCP4 suite[49]. The structure of 17beta-Hydroxysteroid dehydrogenase (PDB entry: 3IS3) was used as the starting model. Coot[50] and REFMAC[51] were used to refine the structure. Atomic coordinates and structure factors have been submitted to the Protein Data Bank (PDB) with the accession numbers 7YB1, 7YB2, 8HFJ and 8HFK (Cartesian coordinates for the atom are provided in Supplementary Data 1).

### Enantioselectivity analysis of CbAR

The absolute configuration of 17β-estradiol and its analogues was determined by comparison with previously reported ¹H NMR data due to the diastereomers formed after the reduction of the carbonyl groups. The enantioselectivity of the purified enzyme over other substrates, except **1l**, was analyzed by chiral HPLC. The enantioselectivity of the purified enzyme over substrate **1l** was analyzed by chiral GC using a chiral GC-column (Betadex-120). The conditions are listed in Supplementary Table 8 and Supplementary Table 9.

### Synthesis of substrates

2-[(17-Oxoestra-1,3,5(10)-trien-3-yl) oxy] acetonitrile (**1c**): Under nitrogen atmosphere, estrone (0.3 g, 1.11 mmol) was dissolved in anhydrous tetrahydrofuran (THF, 5 mL), followed by the addition of sodium methoxide (0.18 g, 3.11 mmol). The bromoacetonitrile (0.66 g, 5.55 mmol), which had been dissolved in 2 mL of THF, was then slowly placed into the reaction tube over a period of 5 min. It was brought to room temperature and stirred for 2 h. After that, the saturated ammonium chloride solution was used to quench the process. Following this, 20 mL of water was added to dilute it before the extraction by EtOAc (3 × 15 mL). The mixed organic layers were washed with 15 mL water for twice, and 15 mL brine. The organic layer was dried over Na₂SO₄, concentrated, and the residue was purified by flash chromatography on silica gel (5% EtOAc in petroleum ether to 20% EtOAc in petroleum ether) to afford 0.27 g (79% yield) **1c** as a white solid.

Estrone acetate (**1d**): The reaction mixture containing estrone (0.3 g, 1.11 mmol), pyridine (0.45 mL, 438 mg, 5.55 mmol), 4-dimethylaminopyridine (DMAP) (13.4 mg, 0.11 mmol) and CH₂Cl₂ (5 mL) was stirred at 0 °C, and acetic anhydride (0.32 mL, 340 mg, 3.33 mmol) was added dropwise. The mixture was then brought to room temperature and stirred overnight. Finally, a saturated ammonium chloride solution was used to quench the process. Following this, 20 mL water was added to dilute it before the extraction by EtOAc (3 × 15 mL). The mixed organic layers were washed with 20 mL water, 30 mL HCl (1M) and 15 mL brine. The organic layer was dried over

Na₂SO₄, concentrated, and the residue was purified by flash chromatography on silica gel (20% EtOAc in petroleum ether to 40% EtOAc in petroleum ether) to afford 0.25 g (72% yield) **1d** as a white solid.

2,2-disubstituted 1,3-cyclodiketones (**1e**, **1f**, **1g**, **1h**, **1i** and **1j**): In a 25 mL flame-dried round bottom flask containing 1N NaOH (1 equiv.), 2-methyl-1,3-cyclopentanedione or 2-methyl-1,3-cyclohexanedione (5 mmol, 1 equiv.) was added and agitated at room temperature for 30 min. Then, the substituted benzyl bromide (10 mmol, 2 equiv.) was added, and the mixture was stirred for another 48 h. Finally, the reaction mixture was extracted with EtOAc (3 × 15 mL) and the mixed organic layers were washed with water (20 mL), 1M HCl (3 × 20 mL) and saturated aqueous NaCl (20 mL). The organic layer was dried over Na₂SO₄, concentrated, and the residue was purified by flash chromatography on silica gel to obtain the dione products[12].

2-bromo-1-(4-bromo-2-hydroxyphenyl)ethan-1-one (**1o**): Cupric bromide (2.23 g, 10 mmol) and 1-(4-bromo-2-hydroxyphenyl) ethanone (1.075 g, 5 mmol) were dissolved in 10 mL of a mixture of CHCl₃/EtOAC (1:1) and placed to reflux at 62 °C for 12 h. The cuprous bromide was filtered using kieselguhr. The filtrate was concentrated in vacuo and the resulting residue was then redissolved in EtOAc. The organic layer was washed twice with brine and dried with Na₂SO₄. Finally, the solvent was removed in vacuo and the product was purified by silica gel chromatography[52].

### Scale-up preparation and characterization of reduced products

Similar to enzymatic activity assay section, 4 mL mixture, instead of 100 μL, was used to prepared the related reduce products. As for substrate emodin, the reaction tube was stirred under nitrogen atmosphere at 25 °C for 12 h, and pure reduced products was purified from the crude product, which was extracted from the reaction solution with EtOAc, by semi-preparative HPLC (Angilent, ZORBAX Eclipse XDB C18 column, 9.4 × 250 mm, 5 μm) with mobile phase (60% acetonitrile and 40% water with 0.1% formic acid (v/v)) and eluted over 20 min at a flow rate of 3 mL/min. As for substrates 1,3-cyclodiketones and estrones, the reaction solution was extracted with EtOAc and the organic solvent was subsequently removed under reduced pressure. The crude product was purified by column chromatography and eluted by EtOAc/petroleum ether mixtures (1:50 to 1:2, v/v) to afford pure reduced products. As for substrates α-haloacetophenones, the reaction solution was extracted with EtOAc and the organic solvent was subsequently removed under reduced pressure. The crude product was further purified by semi-preparative HPLC (Angilent, ZORBAX Eclipse XDB C18 column, 9.4 × 250 mm, 5 μm).

### Reporting summary

Further information on research design is available in the Nature Portfolio Reporting Summary linked to this article.

## Data availability

The atomic structure coordinates of CbAR-NADP⁺, CbAR-NADP⁺-Emodin, CbAR-H162F-NADP⁺−**1e** and CbAR-H162F-NADP⁺-**1o** were deposited in the Protein Data Bank (PDB) (https://www.rcsb.org) under the accession number 7YB1, 7YB2, 8HFJ and 8HFK, respectively. General methods, experimental procedures, characterization of substrates and products, enantioselectivity analysis, ¹H and ¹³C NMR analysis are available within the article and the Supplementary Information. Source data are provided with this paper. Other relevant data are available from the corresponding author on requests. Source data are provided with this paper.

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

## Acknowledgements

We thank the staffs of beamline BL18U1 and BL19U1 of Shanghai Synchrotron Radiation Facility for assistance during X-ray data collection. This work was supported by the National Key R&D Program of China (2021YFC2102700), National Natural Science Foundation of China (32270082), the Natural Science Foundation of Jiangsu Province (BK20202002) and Postgraduate Research & Practice Innovation Program of Jiangsu Province (KYCX20_1812 and KYCX20_1816).

## Author contributions

Y.R. supervised and designed the project; X.H., H.X., Z.Y., Z.D., K.F., Y.G., C.L., and Y.Z. performed research and data analysis; Z.Y. and Y.Z. contribute to data analysis; Y.R. wrote the paper.

## Competing interests

The authors declare no competing interests.
