## [Peer Review File · Nature Communications]

Structural analysis of an anthrol reductase inspires enantioselective synthesis of enantiopure hydroxycycloketones and β -halohydrinsREVIEWER COMMENTS

Reviewer #1 (Remarks to the Author):

This paper discusses the characterisation of an anthrol reductase, some mutagenesis studies and discussion of its crystal structure. While this is an interesting piece of work this reviewer feels it lacks novelty for publication in Nature Communications and would be more suited to a more specialised journal. There are a number of specific comments and questions which would need to be addressed prior to publication.

Are the natural substrates for this class of enzyme known?

Figure 1c traces need a little more information, control vs assay?

Page 4 line 84 supp fig 1d, 1f. Page 4 line 85 supp fig 1e.

Figure 3 a and b orientation of structure makes it very difficult to see what is happening is there a better rotation that could be used and alternative colouring. Zooming in and excluding the magenta prolines may help?

Any mutations to confirm that the catalytic triad functions as expected? Why were mutations not considered to those that were observed by sequence conservation analysis?

Labelling of Table 1 should be more specific to indicate that kinetic parameters relate to 1e

What are the conversions of the wild type enzymes with substrate 1f- 1h? Especially given that H162F had no detectable activity with Emodin?

Crystal structure of 1e with wild type attempted?

Page 10 line 221, typo CbAR-Y201A should read CbAR-Y210A

Testing of further substrates to understand the limitations of the enzyme would greatly improve the paper.

Reviewer #2 (Remarks to the Author):

Review of Structural analysis of an anthrol reductase to inspire an efficient reductive desymmetrization of bulky 1,3-cyclodiketones by Hou et al.

The manuscript reads well and I hope my comments and questions help the authors refine it further.

Starting in the abstract, and also repeated further in the manuscript, the authors state that the "Tyr210 is critical for substrate recruitment and binding as it forms the hydrogen-bond interaction with His162". The authors then go on to describe engineered mutants where His162 is mutated to Phe, which cannot form hydrogen bonds with the Tyr210 residue and these are even more active than the wild type enzyme (at least for some substrates). From the data presented, the Tyr210 does seem to be important, but the hydrogen bond to His162 may not be necessary for substrate binding. In the modelling studies, the Tyr210 is in a different position and does not make any interaction with residue 162. The authors may want to change this statement to something like 'Tyr210 is critical for binding emodin' to avoid confusion with their readers.

Line 55: the 'are' should be changed to 'were'

Line 74: how is "virtual sequence homology screening" different from sequence homology screening?

Line 85: "the melting temperature for CbAR was determined to be 40.27 °C by circular dichroism." I don't think this technique is accurate to a hundredth of a degree, so the authors may want to round to

40.3 C.

Line 102: "its catalytic mechanism has not been investigated yet." Should probably read "its catalytic mechanism had not been investigated."

Line 104: "It showed that CbAR folds as a tetramer" – I don't believe this is true. Supp. figure 2 shows four molecules in the asymmetric unit but the contacts made look like crystallographic interfaces, not oligomeric interfaces forming a tetrameric structure. The authors should check this (there are various software packages including PISA which may help) and also check by gel filtration or some other technique to show tetramer formation in solution if they think this really is the case. And if it is the case, they should have a figure to show this (replacing Supp. figure 2).

Line 112: "indicating that this region plays a critical role in substrate recruitment and binding." I agree that this region is important for binding emodin, but the authors haven't shown any data to prove it is important for recruitment of substrates.

Lines 130-131 and figure 3c: I'm not clear how Lys169 forming hydrogen bonds to the ribose ring of NADPH reduces the pKa of Tyr165. The authors cite reference 33, but this is performing the opposite reaction, oxidation instead of reduction, and in figures 3b and 3d, the lysine nitrogen looks to be pointing away from the Tyr165 hydroxyl group. Oxidation and reduction can be done with the same enzyme running the other direction, but one might expect that the parameters (like pH) might also work the opposite direction to accomplish this. Maybe the authors can explain this better or provide a view of the active site which makes this clearer to the reader.

Line 136: "are hydrogen" should be "is hydrogen"

Line 141: "we concluded that residues Gln152, His162 and Tyr210 play critical" should read "we conclude that residues Gln152, His162 and Tyr210 play critical". In figures 3a-3c, residue 152 is labelled as Asn- is it Asn or Gln?

Lines 156-170: The authors analyse their mutant enzyme data and suggest that residues in the loop over the active site, 209-216, play an important role in recruitment of substrate. I can see how these residues may stabilize the substrate and hold it in place for the reaction to occur, but would like to know how these residues recruit the enzyme from bulk solution into the active site? I'm not familiar with this concept and would like it explained in more detail. In the next section, engineering the enzyme to accept a different substrate, they mutate the same residues (His162Phe and Tyr210Phe) that previously were discussed as being absolutely required for activity, 'recruitment' and binding of substrate. I suggest that prior to the engineering section (lines 171 onward) that the authors make it clear that the enzymatic (WT and mutants) data only refers to emodin by replacing 'substrate' with emodin as the data presented in this first part of the paper only apply to emodin and not substrates more generally (as shown by their engineering and modelling work).

Line 178: Drop "Then," from the start of the sentence or change it to "We then used CbAR as ...".

Line 186: It isn't clear to me that 1a and 1e are shown in Supp figure 1e and 1f. Supp figure 1e and 1f only refer to emodin as far as I can tell.

Lines 186 and 193: I still disagree with the authors regarding the recruitment abilities of the loop/ Tyr210.

Lines 197 to 203: The authors state "The results suggest that the loop region 209-216 located at the entrance of the active pocket could be used as the potential target for protein engineering to further expand the substrate range of CbAR." But the results and Table 2 are all done with His162Phe, which is not in the loop 209-216, so there seems to be a disconnect between the data and the discussion.

Line 211: 'modulein' should be 'module in'

Line 214: 'the reactive carbonyl groups' should read 'the reactive carbonyl group'

Line 215: 3.3 and 3.4 Angstrom hydrogen bonds are rather long for this purpose.

Line 218: I still don't agree with the 'recruit' statement.

Conclusions: I suggest that the strong hydrogen bond between His162 and Tyr210 hypothesis should be tested more thoroughly. Changing His162 to something other than Phe, which may cause steric clashes with Tyr210 and not allow the loop to shut over emodin, would make a stronger case for their hypothesis (mutations such as Ala to remove potential hydrogen bonding without causing a steric clash and something like Asn to change the residue but keep potential hydrogen bonding capability).

Lines 253-255: I agree that their data show good stereochemical product selectivity for the substrates they tested, but these lines suggest that there is selectivity for substrates as well, for which data are not presented. Rewording this to make it clearer to the readers what has been shown is important.

Supplementary table 1- X-ray statistics:

I find it hard to believe that the B-factors for a 3.3 Angstrom structure are lower than those found in the 1.85 Angstrom structure and these B-factors don't agree with what is found in the PDB reports. Speaking of the PDB reports, I find it remarkable that the emodin and NADP structures found in these reports are 'perfect'. There are no outliers or strained torsions, rings, bond angles and bond lengths- suggesting that these may not have been refined with the protein structures. This is also a little strange given that the structures were not refined with particularly tight constraints on the bond lengths or angles (see Supp Table 1 statistics regarding bond lengths and angles).

It is common to show difference density maps of substrates/ligands/cofactors found in the active site of enzymes and I suggest this would be a good addition to the manuscript so readers can see the density for themselves.

The authors have put together a nice manuscript describing their work and it is well written. I disagree with a few of the statements they've made regarding 'recruitment' of substrate into the active site and the importance of the hydrogen bond between His162 and Tyr210, but with further experiments they can prove my intuition incorrect. It would be good to show the density in the active site, particularly difference density for the emodin as well. It would also be good to have them check Table 1 X-ray statistics to make sure there are no typos and that they more closely align with the PDB reports.

Reviewer #3 (Remarks to the Author):

Hou et al reported the identification and characterization of a new anthrol reductase from fungi *Cercospora*, (CbAR). The enzyme has been shown to catalyze the reduction of substrates (emodin and estrones and analogues) using NADPH. The reduction of the same substrates had also been shown in earlier studies where anthrol reductases such as MdpC, AfIM, Arti or ARti-2 were used. To understand the mechanism of emodin hydroquinone by CbAR, they obtained the crystal structure co-crystallized with NADPH along with or without emodin, which gave vital information about the binding site and active site of the enzyme. The analysis of crystal structures of CbAR with and without emodin revealed the role of amino acid residues 209-216 in substrate recruitment. In addition, Gln152, His162 and Tyr210 have been shown to play crucial role in substrate recruitment and binding. This is a significant finding to understand the mechanism of functioning of this new class of enzymes, the anthrol reductases.

Authors also predict the possible reduction of 2,2-disubstituted 1,3-cyclodiketones by CbAR to obtain challenging 2,2-disubstituted-3-hydroxycycloketones as products. Accordingly, 2,2-disubstituted 1,3-cyclodiketones were reduced to obtained (2S, 3S)-ketols with high conversion and enantiomeric access. With the help of docking studies, they identified the key amino acid residues to further enhance the activity and there prepared mutants. The use of CbAR-H216F gave the corresponding (2S,3S)-ketodiols with >99% conversion. The CbAR-H162F mutant gave 44.7 fold increase compared to WT enzyme. These results suggests that further manipulation of 209-216 region may allow the reduction of other substrates by CbAR.

Overall it is a nice piece of work, where authors could give insight into the functioning of anthrol reductase CbAR for the first time. The binding of emodin at the at the active site gives crucial information about the reduction of anthrols at the active site. The reduction of 2,2-disubstituted 1,3-cyclodiketones to obtained (2S, 3S)-ketols is also a significant result. The work is of high significance to the field of biocatalysis. The data supports the findings of the work.

Based on the assessment and the work done I recommend the publication of the work in Nature Communications after revision as mentioned below.

Comments:

Page 1, line 1: Please mention the same title in manuscript and SI.

Page 2, line 16-19: Sentence "Then, crystal....." is too long and confusing. It is better to rephrase it.

Page 2, line 21: The conversion rates can be written as 93% instead of 92.9%.

Page 3, line 37-40: The sentence "However, enzymes....." is too long and confusing. Please

replace it or divide into two sentences.

Page 3, line 40: Use "this is because" instead use "The reason is that"

Page 4, line 70: please make it 93% instead of 92.9%. Apply same throughout the manuscript.

Page 4, line 75: "the SDR superfamily and shares the highest sequence identity of 86% with 17 β -HSDcl. Please also give sequence identity with other anthrol reductases. Can be include in SI.

Page 4, line 81: please modify "conversion rate" to "conversion".

Page 4, line 85: the temperature is written as 40.27 °C. Make it up to one decimal places

Page 4, line 89: instead of "could" better to use "is known to catalyze"

Page 4, line 89-91: The sentence is not clear. Please rephrase the sentence or divide the content into two sentences.

Page 4, line 92-93: The inference drawn from the reduction of estrone and its analogues (1b, 1c, 1d) by CbAR to reduce 2,2-disubstituted 1,3-cyclodiketones is not very convincing. The authors have not look at the possibility of other related substrates. Although, 17 β -HSD of *Curvularia lunata* has been shown to catalyze the reduction of estrones, its physiological function is more likely to be the reduction of anthrols.

Why authors have not considered six membered 1,3-diketones for reduction by CbAR. It is better to provide further justification for choosing 2,2-disubstituted 1,3-cyclodiketones as the substrates.

Page 8, line 172: "owing to two chiral centers of 2,2-disubstituted 1,3-cyclodiketones" need to be corrected.

Page 8, line 174-178. The sentence "Since, CbAR can accommodate....." is confusing and too long. please rephrase it.

Page 8, line 182: "which could be used for the preparation of many natural products" how? If the authors can give the structures of the natural products such as such as cortistatin A, clavulactone, madindoline A, digitoxigenin and hygrophorone by highlighting the 2,2-disubstituted-3-hydroxycycloketones moiety, it will be helpful to understand the significance of the compounds that have been synthesized. This information may be included either in the manuscript or Supporting Information.

Page 9, line 200: replace 92.9% with 93%

Page 14, line 335: The absolute configuration of 17 β -esterdiol and its analogues was determined by comparison of previously reported ¹H NMR spectra. This statement is incorrect, please correct it.

S24-S37: The number and structures on the NMR spectra are not clearly visible. Please increase the font size and put structures in high resolution.

Response to reviewers:

Reviewer 1: This paper discusses the characterisation of an anthrol reductase, some mutagenesis studies and discussion of its crystal structure. While this is an interesting piece of work this reviewer feels it lacks novelty for publication in Nature Communications and would be more suited to a more specialised journal. There are a number of specific comments and questions which would need to be addressed prior to publication.

A: We would like to thank your professional feedbacks and constructive suggestions of this manuscript, which help us improve the quality of this manuscript.

At first, in this study, although the reduction of emodin by an anthrol reductase was realized in 2012, the structure-function relationship between anthrol reductase and emodin was revealed for the first time. Based on the crystal structure of CbAR-Emodin complex, the stereoselectivity of the reduction of emodin is well explained. Secondly, as you suggested, we explored further substrates to understand the limitations of the enzyme, which greatly improves this paper. To our delight, besides diverse 2,2-disubstituted 1,3-cyclopentadiketones, six membered 1,3-diketone and α -haloacetophenones could also be reduced to optically pure (2*S*, 3*S*)-2-benzyl-3-hydroxy-2-methylcyclohexan-1-one and (*R*)- β -halohydrins by variant CbAR-H162F after structure-guided engineering (Fig. 4, 5), respectively. Last, their solid stereoselectivity mechanisms were well explained by the respective new crystal structures of CbAR-H162F-substrate complex (Fig. 4, 5), which was also proposed by you. In previous version, this was explained by molecular dynamics (MD) simulations and molecular docking. New Fig 4 and 5 were provided to illustrate our new results and findings as follow. Therefore, we believe that the revised version for the characterization and well-explained catalytic mechanism of CbAR with three CbAR-substrate crystal structures now possess novelty of this study, and is suitable for publication after addressing all your comments.

Fig. 4 Molecular insights of reductive desymmetrization of 2,2-disubstituted prochiral 1,3-cyclodiketones for the preparation of (2*S*, 3*S*)-2,2-disubstituted-3-hydroxycycloketones by CbAR-H162F. (a) Four possible stereoisomers generated by reduction desymmetrization of 2,2-disubstituted-1,3-cyclodiketones. **1e–1j** represent 6 prochiral 1,3-cyclodiketones used in this study. (b) Reduced compounds (**2e–2j**) obtained by the reduction of substrates (**1e–1j**) by CbAR-H162F. (c) Substrate binding pocket of CbAR-H162F for substrate **1e**. The 2Fo – Fc electron density maps of substrate **1e** was contoured at 1.0 σ in blue color (PDB: 8HFJ). (d) Proposed proton shuttling mechanism and substrate binding mode of CbAR-H162F for the reduction of substrate **1e**. The proposed proton shuttling mechanism of CbAR towards substrate **1e** is indicated by the blue arrow.

Fig. 5 Molecular insights of asymmetric reduction of α -haloacetophenones for the preparation of β -halohydrins by CbAR-H162F. (a) Two possible stereoisomers generated

by reduction of α -haloacetophenones. **1k-1o** represents 5 α -haloacetophenones used in this study. (b) Reduced compounds (**2k-2o**) obtained by the reduction of substrates (**1k-1o**) by CbAR-H162F. n.d. indicates not detected. (c) Substrate binding pocket of CbAR-H162F for substrate **1o** (PDB: 8HFK). Atoms S and Br are colored in yellow and green, respectively. The 2Fo – Fc electron density maps of substrate **1o** and Met202 was contoured at 1.0 σ in blue color. (d) Proposed proton shuttling mechanism and substrate binding mode of CbAR-H162F for the reduction of substrate **1o**. The proposed proton shuttling mechanism of CbAR towards substrate **1o** is indicated by the blue arrow.

Q1: - Are the natural substrates for this class of enzyme known?

A1: Thank you very much for your question! Typically, the natural substrate for anthrol reductase is emodin or its analogue. Among several anthrol reductases we have listed in our manuscript, emodin is the natural substrate for all the enzymes except for AfIM, whose natural substrate is the emodin analogue versicolorin A. Thus, we have added the corresponding description into the revised manuscript as follows.

For example, emodin hydroquinone, as the natural substrate of most anthrol reductases, can be reduced to (*R*)-3,8,9,10-tetrahydroxy-6-methyl-3,4-dihydroanthracene-1(2*H*)-one (**2a**), a key intermediate for the biosynthesis of many natural products, such as agnestin A²⁰, ravenelin²¹, shamixanthone²², secalonic acids²³, (-)-flavoskyrin²⁴, and hemi-cryptosporioptide²⁵ (Fig. 1a).

Q2:- Figure 1c traces need a little more information, control vs assay?

A2: Thank you very much for nice advice! We have added the control vs assay and line legends in Fig. 1c to make this figure more informative as you suggested.

Fig. 1 Identification of anthrol reductase CbAR with the ability to catalyze the reduction of

emodin and estrones. (a) The reduction of emodin catalyzed by anthrol reductases (ARs) in the presence of NADPH and Na₂S₂O₄. The reduced product **2a** acts as an important intermediate for the biosynthesis of many natural products. (b) Phylogenetic tree analysis of CbAR. (c) HPLC profiles of CbAR catalyzed the reduction of emodin in the presence of NADPH and Na₂S₂O₄. (d) Estrone and its analogues could be reduced by CbAR. The reactive positions are highlighted.

Q3:- Page 4 line 84 supp fig 1d, 1f. Page 4 line 85 supp fig 1e.

A3: Thank you very much for your correction! We have corrected the citation of previous supp Fig. 1d, e, and f to supp Fig. 1f, g, h due to the addition of two new figures in the revised ESI. In addition, we have checked all cites of tables and figures in the revised manuscript and ESI to avoid similar mistakes.

Q4: - Figure 3a and b orientation of structure makes it very difficult to see what is happening is there a better rotation that could be used and alternative colouring. Zooming in and excluding the magenta prolines may help?

A4: Thank you very much for nice advice! We removed the rotation view and added the 2Fo – Fc electron density maps of emodin in Fig 3b. In addition, according to your suggestion, new supplementary Fig. 3, which was zoomed in and excluded the magenta prolines on the basis of Fig. 3b, was added in the supplementary materials.

Supplementary Figure 3. Close-up views of substrate binding pocket of CbAR with emodin. (a) NADP⁺, the catalytic triad (Ser151, Tyr165 and Lys169) and residues involved in binding of emodin (N152, H162 and Y210) are highlighted in different colors. The 2Fo – Fc electron density maps of emodin was contoured at 1.0 σ in blue color. The proposed proton shuttling mechanism of CbAR towards the substrate is indicated by the blue arrow. (b) Close-up views of a 120° rotation along the X-axis of (a). The distance between Lys169, NADP⁺ and Tyr165 were highlighted in yellow color.

Q5: - Any mutations to confirm that the catalytic triad functions as expected? Why were mutations not considered to those that were observed by sequence conservation analysis?

A5: Thank you very much for your professional advice. Based on the amino sequence conservation analysis, site-directed mutagenesis was performed on amino acids involved

in catalytic triad. As expected, these variants completely abolished the catalytic activity (Fig. 3d). These results indicated the catalytic triad plays an indispensable role during the reaction. Additionally, the gate region from residue 209 to 216 was also investigated through mutation, showing that besides Tyr210, Ile211 is also critical for the catalytic activity of CbAR (Supplementary Fig. 4).

Q6: - Labelling of Table 1 should be more specific to indicate that kinetic parameters relate to **1e**.

A6: Thank you very much for your suggestion. We have changed the table label to "Kinetic parameters of wild-type CbAR and its variants towards 2-methyl-2-(4-methylbenzyl) cyclopentane-1,3-dione (**1e**).", and placed Table 1 in supplementary materials as Supplementary Table 4.

Q7: - What are the conversions of the wild type enzymes with substrate 1f- 1h? Especially given that H162F had no detectable activity with Emodin?

A7: Thank you very much for your suggestions. According to your suggestions, we have tested the conversions and ee values of the wild-type enzyme with substrates **1f-1j**, and the results are shown in Supplementary Table 3. It shows that all of them could be efficiently converted into the corresponding (2*S*, 3*S*)-ketols (**2e-2i**) with higher than 93% conversion and 99% ee by CbAR-H162F, whose conversions were better than that of wild-type CbAR.

Supplementary Table 3. Asymmetric reduction of various 2,2-disubstituted-1,3-cyclodiketones by CbAR and CbAR-H162F variant.

Substrate	Enzymes	Conversion (%)	Stereoisomeric distribution of the ketol product (%)			
			(2 R , 3 S)	(2 R , 3 R)	(2 S , 3 S)	(2 S , 3 R)
1e	WT	46	-	-	>99	-
	H162F	99	-	-	>99	-
1f	WT	42	-	-	>99	-
	H162F	98	-	-	>99	-
1g	WT	29	-	-	>99	-

	H162F	93	-	-	>99	-
1h	WT	76	-	-	>99	-
	H162F	99	-	-	>99	-
1i	WT	70	-	-	>99	-
	H162F	97	-	-	>99	-
1j	WT	19	-	-	>99	-
	H162F	46	-	-	>99	-

Q8: - Crystal structure of 1e with wild type attempted?

A8: Thank you very much for your great suggestions. We have attempted to obtain the crystal structure of CbAR-substrate (**1e-1j**) or CbAR-H162F-substrate (**1e-1j**) complex through co-crystallization and soaking. Fortunately, we have obtained the crystal structure of CbAR-H162F-**1e** complex with 2.75 Å resolution (PDB code: 8HFJ). The 2-methyl-cyclopentane-1,3-dione part of **1e** is clearly observed and well localized to the active site of CbAR-H162F (Fig. 4c, d). Two carbonyl groups of **1e** form the hydrogen-bond interactions with the amino group of Gly197 and the hydroxyl group of Tyr165 (Fig. 4c, d), respectively. The interaction between Tyr165-OH and **1e** facilitates hydride transfer for the reduction. Additionally, it shows that the 2-methyl group of **1e** orients towards co-factor NADP⁺ (Fig. 4c). This helps us better reveal the exclusive stereoselectivity for 2,2-disubstituted 1,3-cyclodiketones by CbAR-H162F.

Fig. 4 Molecular insights of reductive desymmetrization of 2,2-disubstituted prochiral 1,3-cyclodiketones for the preparation of (2*S*, 3*S*)-2,2-disubstituted-3-hydroxycycloketones by CbAR-H162F. (a) Four possible stereoisomers generated by reduction desymmetrization of 2,2-disubstituted-1,3-cyclodiketones. **1e-1j** represent 6 prochiral 1,3-cyclodiketones

used in this study. (b) Reduced compounds (**2e–2j**) obtained by the reduction of substrates (**1e–1j**) by CbAR-H162F. (c) Substrate binding pocket of CbAR-H162F for substrate **1e**. The 2Fo – Fc electron density maps of substrate **1e** was contoured at 1.0 σ in blue color (PDB: 8HFJ). (d) Proposed proton shuttling mechanism and substrate binding mode of CbAR-H162F for the reduction of substrate **1e**. The proposed proton shuttling mechanism of CbAR towards substrate **1e** is indicated by the blue arrow.

Q9: - Page 10 line 221, typo CbAR-Y201A should read CbAR-Y210A.

A9: Thank you very much for your careful correction! We have recently solved the crystal structure of CbAR-H162F-**1e** complex. Hence, we added the more convincing results of the crystal structure to replace the relevant content of Molecular Docking and Molecular Dynamics Simulations in our revised manuscript.

Q10: - Testing of further substrates to understand the limitations of the enzyme would greatly improve the paper.

A10: Thank you very much for your excellent suggestions. According to your suggestions, we have further explored the substrate scope of CbAR and CbAR-H162F variant to 2,2-disubstituted 1,3-cyclohexanediketone, acetophenone and α -haloacetophenones. To our delight, in addition to the reduction of 2-benzyl-2-methylcyclohexane-1,3-dione (**1j**) to optically pure (2*S*, 3*S*)-ketol (**2j**), CbAR-H162F variant could also convert α -haloacetophenones (**1i–1o**) to optically pure (*R*)- β -halohydrins (**2i–2o**) (Fig. 4a, 5). This indicates that the engineered CbAR-H162F is a versatile reductase for a broad range of prochiral ketones. Furthermore, the crystal structure of CbAR-H162F/ α -haloacetophenone complex was studied to understand the excellent enantioselectivity for α -haloacetophenones. Therefore, these results greatly improve our manuscript as you suggested.

Fig. 5 Molecular insights of asymmetric reduction of α -haloacetophenones for the

preparation of β -halohydrins by CbAR-H162F. (a) Two possible stereoisomers generated by reduction of α -haloacetophenones. **1k-1o** represents 5 α -haloacetophenones used in this study. (b) Reduced compounds (**2k-2o**) obtained by the reduction of substrates (**1k-1o**) by CbAR-H162F. n.d. indicates not detected. (c) Substrate binding pocket of CbAR-H162F for substrate **1o** (PDB: 8HFK). Atoms S and Br are colored in yellow and green, respectively. The $2F_o - F_c$ electron density maps of substrate **1o** and Met202 was contoured at 1.0σ in blue color. (d) Proposed proton shuttling mechanism and substrate binding mode of CbAR-H162F for the reduction of substrate **1o**. The proposed proton shuttling mechanism of CbAR towards substrate **1o** is indicated by the blue arrow.

Reviewer 2: Review of Structural analysis of an anthrol reductase to inspire an efficient reductive desymmetrization of bulky 1,3-cyclodiketones by Hou et al. The manuscript reads well and I hope my comments and questions help the authors refine it further.

A: We appreciate your positive evaluations and valuable suggestions, which help us improve the quality of this manuscript. Here we prepared this point-to-point response and highlighted the changes in blue.

Q1: Starting in the abstract, and also repeated further in the manuscript, the authors state that the “Tyr210 is critical for substrate recruitment and binding as it forms the hydrogen-bond interaction with His162”. The authors then go on to describe engineered mutants where His162 is mutated to Phe, which cannot form hydrogen bonds with the Tyr210 residue and these are even more active than the wild type enzyme (at least for some substrates). From the data presented, the Tyr210 does seem to be important, but the hydrogen bond to His162 may not be necessary for substrate binding. In the modelling studies, the Tyr210 is in a different position and does not make any interaction with residue 162. The authors may want to change this statement to something like ‘Tyr210 is critical for binding emodin’ to avoid confusion with their readers.

A1: Thank you very much for your great advice. You are right, based on new crystal structures of CbAR-H162F-substrate (including 2,2-disubstituted 1,3-cyclodiketones and α -haloacetophenones), Tyr210 is only critical for emodin recognition and binding, but not for other substrates. As suggested, this statement is now rewritten as “It reveals that Tyr210 is critical for emodin recognition and binding, as it forms hydrogen-bond interaction with His162 and π - π stacking interaction with emodin, respectively” and highlighted in red color.

Q2: Line 55: the ‘are’ should be changed to ‘were’

A2: Thank you very much for your correction! We have changed the “are” to “were” in the revised manuscript and highlighted in red color.

Q3: Line 74: how is “virtual sequence homology screening” different from sequence homology screening?

A3: Thank you very much for your correction! Virtual sequence homology screening is a method to screen potential interacting molecules for target protein. Sequence homology screening is the most widely used, and most reliable strategy for characterizing newly determined sequences. In this manuscript, “sequence homology screening” is more appropriate and we have rewritten “virtual sequence homology” as “sequence homology screening” in the revised manuscript. Thanks again for your correction.

Q4: Line 85: “the melting temperature for CbAR was determined to be 40.27 °C by circular dichroism.” I don’t think this technique is accurate to a hundredth of a degree, so the authors may want to round to 40.3 °C.

A4: Thank you very much again for your correction. As suggested, we have changed 40.27 to 40.3°C in our manuscript and Supplementary Fig. 1g and 1h.

Q5: Line 102: “its catalytic mechanism has not been investigated yet.” Should probably read “its catalytic mechanism had not been investigated.”

A5: Thank you very much for your suggestions. We have rewritten “its catalytic mechanism has not been investigated yet.” as “its catalytic mechanism had not been investigated.” and highlighted it in red in the manuscript.

Q6: Line 104: “It showed that CbAR folds as a tetramer” – I don’t believe this is true. Supp. figure 2 shows four molecules in the asymmetric unit but the contacts made look like crystallographic interfaces, not oligomeric interfaces forming a tetrameric structure. The authors should check this (there are various software packages including PISA which may help) and also check by gel filtration or some other technique to show tetramer formation in solution if they think this really is the case. And if it is the case, they should have a figure to show this (replacing Supp. figure 2).

A6: Thank you very much again for your wonderful suggestions. Based on the result of size-exclusion chromatography (SEC), CbAR exist as a dimer in solution. Gel filtration was conducted, and the results were provided as Supplementary Fig. 1b and c in the supporting information. Meanwhile, we also used the PISA server to analyze the surface area and assemblies of CbAR as suggested. It was found that the buried surface area between two subunits is 1844 Å², corresponding to 15.0% of the accessible surface area of one monomer. This result also indicated CbAR exist as a dimer in solution.

Therefore, we have corrected the corresponding description to “It showed that CbAR folds as a tetramer (Supplementary Fig. 2), but it was eluted as a dimer in solution (Supplementary Fig. 1b, 1c).” in the revised manuscript.

Supplementary Figure 1. (b) The concentrated CbAR protein (5 mg/mL) eluted as a single peak 73 mL when it was uploaded onto a HiLoad 16/600 Superdex 200 pg gel-filtration column. (c) Standard markers were eluted at different volume from a HiLoad 16/600 Superdex 200 pg gel-filtration column. (1. Myoglobin, 1.5 mg/mL, Mr 17 kDa. 2. Ovalbumin, 5 mg/mL, Mr 44 kDa. 3. Albumin, 5 mg/mL, Mr 66 kDa. 4. IgG, 0.2 mg/mL, Mr 158 kDa. 5. Ferritin, 0.24 mg/mL, Mr 440 kDa.)

Q7: Line 112: “indicating that this region plays a critical role in substrate recruitment and binding.” I agree that this region is important for binding emodin, but the authors haven’t shown any data to prove it is important for recruitment of substrates.

A7: Thank you very much again for your wonderful suggestions. As suggested, site-directed mutagenesis for His209-Thr216 region was carried out to validate the importance of these residues. As shown in Supplementary Fig. 4, all of the mutants tested showed impaired activity compared to wild-type. In particular, the activity of Y210F and I211A were completely abolished. These results indicated His209-Thr216 region plays an important role in the reduction of emodin.

Supplementary Figure 4. The relative activity of wild-type (WT) CbAR and its variants towards emodin. G214 was not mutated in this study owing to the small side chain similar to alanine. ND, not detected. All experiments were carried out in triplicate, and error bars indicate \pm sd.

Q8: Lines 130-131 and figure 3c: I'm not clear how Lys169 forming hydrogen bonds to the ribose ring of NADPH reduces the pKa of Tyr165. The authors site reference 33, but this is performing the opposite reaction, oxidation instead of reduction, and in figures 3b and 3d, the lysine nitrogen looks to be pointing away from the Tyr165 hydroxyl group. Oxidation and reduction can be done with the same enzyme running the other direction, but one might expect that the parameters (like pH) might also work the opposite direction to accomplish this. Maybe the authors can explain this better or provide a view of the active site which makes this clearer to the reader.

A8: Thank you very much for your suggestions. The catalytic triad in the reduction of carbonyl group catalyzed by SDR family enzymes has been well documented, and the conserved Lys-residue has been reported to play a dual role in the catalytic triad, which orients the cofactor by forming hydrogen bonds to the nicotinamide-ribose moiety, and lowers the pKa of the catalytic Tyr via electrostatic interaction (Hoffmann, et al, 2007, *Drug Metab. Rev.*). We regret for any confusion caused by our imprecise language. The inappropriate statement has been rewritten as “Ser151 plays a role in stabilizing the substrate. Tyr165 serves as a general base for proton transfer. Lys169 can not only form the hydrogen-bond interaction with NADPH but also reduce the pKa of Tyr-OH.” In addition, we have provided a clear view of the active site in Supplementary Fig. 3, as suggested.

Supplementary Figure 3. Close-up views of substrate binding pocket of CbAR with emodin. (a) NADP⁺, the catalytic triad (Ser151, Tyr165 and Lys169) and residues involved in binding of emodin (N152, H162 and Y210) are highlighted in different colors. The 2Fo – Fc electron density maps of emodin was contoured at 1.0 σ in blue color. The proposed proton shuttling mechanism of CbAR towards the substrate is indicated by the blue arrow. (b) Close-up views of a 120° rotation along the X-axis of (a). The distance between Lys169, NADP⁺ and Tyr165 were highlighted in yellow color.

Q9: Line 136: “are hydrogen” should be “is hydrogen”

A9: Thank you very much for your correction! We have rewritten “are hydrogen” as “is hydrogen” and highlighted it in red in our manuscript.

Q10: Line 141: “we concluded that residues Gln152, His162 and Tyr210 play critical” should read “we conclude that residues Gln152, His162 and Tyr210 play critical”. In figures 3a-3c, residue 152 is labelled as Asn- is it Asn or Gln?

A10: Thank you very much for your correction of our careless mistake for the residue 152! We have rewritten the sentence as “we conclude that residues Asn152, His162 and Tyr210 play critical roles in substrate recognition, binding and stabilization of emodin” in our manuscript.

Q11: Lines 156-170: The authors analyse their mutant enzyme data and suggest that residues in the loop over the active site, 209-216, play an important role in recruitment of substrate. I can see how these residues may stabilize the substrate and hold it in place for the reaction to occur, but would like to know how these residues recruit the enzyme from bulk solution into the active site? I’m not familiar with this concept and would like it explained in more detail. In the next section, engineering the enzyme to accept a different substrate, they mutate the same residues (His162Phe and Tyr210Phe) that previously were discussed as being absolutely required for activity, ‘recruitment’ and binding of substrate. I suggest that prior to the engineering section (lines 171 onward) that the authors make it clear that the enzymatic (WT and mutants) data only refers to emodin by replacing ‘substrate’ with emodin as the data presented in this first part of the paper only apply to emodin and not substrates more generally (as shown by their engineering and modelling work).

A11: Thank you very much for your wonderful suggestions. We have rewritten “substrate” as “emodin” in our first part prior to the engineering to avoid the misunderstanding and highlighted it in red in our manuscript. As for “recruitment” role of 209-216 region, we also think it is not appropriate after carefully considering your suggestion, hence, we have rewritten “recruitment” as “recognition” in our manuscript and highlighted it in red.

Q12: Line 178: Drop “Then,” from the start of the sentence or change it to “We then used CbAR as ...”.

A12: Thank you very much again for your wonderful suggestions. We have rewritten “Then, CbAR was used as the biocatalyst ...” as “We then used CbAR as the biocatalyst ...” in our manuscript and highlighted it in red.

Q13: Line 186: It isn’t clear to me that 1a and 1e are shown in Supp figure 1e and 1f. Supp

figure 1e and 1f only refer to emodin as far as I can tell.

A13: Thank you very much for your wonderful suggestions. We have revised the manuscript, and cited Supplementary Table 4 for substrate **1e** and Supplementary Fig. 1f, h for substrate emodin (**1a**) to make this clearer. The revised description was attached as follows.

It showed that its K_m and k_{cat} values were 5.13 ± 0.33 mM and 0.58 ± 0.01 min⁻¹ (Supplementary Table 4), respectively, suggesting that the catalytic activity of CbAR towards **1e** is much lower than that towards **1a** (Supplementary Fig. 1f, h), which could be explained by the precise recognition or orientation of **1a** by Tyr210 (Fig. 3b, c), with the formation of the hydrogen-bond interaction with His162.

Q14: Lines 186 and 193: I still disagree with the authors regarding the recruitment abilities of the loop/ Tyr210.

A14: Thank you very much again for your wonderful suggestions. We have changed the description on “recruitment” to “recognition” in our manuscript to avoid the misunderstanding, and highlighted it in red.

Q15: Lines 197 to 203: The authors state “The results suggest that the loop region 209-216 located at the entrance of the active pocket could be used as the potential target for protein engineering to further expand the substrate range of CbAR.” But the results and Table 2 are all done with His162Phe, which is not in the loop 209-216, so there seems to be a disconnect between the data and the discussion.

A15: Thank you very much again for your wonderful suggestions. We have removed this statement in our manuscript and rewritten the description as follows:

All of them could be efficiently converted into the corresponding (2*S*, 3*S*)-ketols (**2e-2i**) with higher than 93% conversion and 99% *ee* values (Fig. 4b and Supplementary Table 3), whose conversions were better than that of wild-type CbAR (Supplementary Table 3). Furthermore, optically pure (2*S*, 3*S*)-2-benzyl-3-hydroxy-2-methylcyclohexan-1-one (**2j**) could also be obtained from the related six membered 1,3-diketone (**1j**) by CbAR-H162F (Fig. 4b).

Q16: Line 211: ‘modulein’ should be ‘module in’

A16: Thank you very much for your correction! We have recently solved the crystal structure of CbAR-H162F-**1e** complex. Hence, we added the more convincing results of

the crystal structure to replace the relevant content of Molecular Docking and Molecular Dynamics Simulations in our revised manuscript.

Q17: Line 214: ‘the reactive carbonyl groups’ should read ‘the reactive carbonyl group’

A17: Thank you very much again for your correction! Due to the addition of the more convincing results of the crystal structure to replace Molecular Docking and Molecular Dynamics Simulations, we have removed the relevant part in our manuscript.

Q18: Line 215: 3.3 and 3.4 Angstrom hydrogen bonds are rather long for this purpose.

A18: Thank you very much for your suggestions. Due to the addition of the more convincing results of the crystal structure (Fig. 4) to replace Molecular Docking and Molecular Dynamics Simulations, we have removed the relevant part in our manuscript.

Fig. 4 Molecular insights of reductive desymmetrization of 2,2-disubstituted prochiral 1,3-cyclodiketones for the preparation of (2*S*, 3*S*)-2,2-disubstituted-3-hydroxycycloketones by CbAR-H162F. (a) Four possible stereoisomers generated by reduction desymmetrization of 2,2-disubstituted-1,3-cyclodiketones. **1e–1j** represent 6 prochiral 1,3-cyclodiketones used in this study. (b) Reduced compounds (**2e–2j**) obtained by the reduction of substrates (**1e–1j**) by CbAR-H162F. (c) Substrate binding pocket of CbAR-H162F for substrate **1e**. The 2Fo – Fc electron density maps of substrate **1e** was contoured at 1.0 σ in blue color (PDB: 8HFJ). (d) Proposed proton shuttling mechanism and substrate binding mode of CbAR-H162F for the reduction of substrate **1e**. The proposed proton shuttling mechanism of CbAR towards substrate **1e** is indicated by the blue arrow.

Q19: Line 218: I still don't agree with the 'recruit' statement.

A19: Thank you very much again for your wonderful suggestions. We have rewritten "recruitment" as "recognition" in our manuscript and highlighted it in red.

Q20: Conclusions: I suggest that the strong hydrogen bond between His162 and Tyr210 hypothesis should be tested more thoroughly. Changing His162 to something other than Phe, which may cause steric clashes with Tyr210 and not allow the loop to shut over emodin, would make a stronger case for their hypothesis (mutations such as Ala to remove potential hydrogen bonding without causing a steric clash and something like Asn to change the residue but keep potential hydrogen bonding capability).

A20: Thank you very much again for your wonderful suggestions. As suggested, we have tested the activity of H162A and H162N variants towards emodin. As shown in Fig. 3d attached as follows, the activities of H162A and H162N towards emodin were also completely abolished. These results indicated His162 plays an indispensable role during reduction of emodin, in addition to hydrogen bonding with Tyr210, His162 may also participated in the substrate binding pocket regulation of CbAR by remote effects to ensure enzyme specificity.

Fig. 3. (d) The relative activity of wild-type (WT) CbAR and its variants towards emodin. ND, not detected. All experiments were carried out in triplicate, and error bars indicate \pm sd.

Q21: Lines 253-255: I agree that their data show good stereochemical product selectivity for the substrates they tested, but these lines suggest that there is selectivity for substrates as well, for which data are not presented. Rewording this to make it clearer to the readers what has been shown is important.

A21: Thank you very much for your suggestions. With the new results and findings (Fig. 5), we have added the statement of the substrate scope in our revised manuscript and relevant content has been refined as follows:

Furthermore, CbAR-H162F could also smoothly reduce various α -haloacetophenone to afford optically pure β -halohydrins with excellent enantioselectivity (Fig. 5). More importantly, their excellent stereoselectivity mechanisms were well revealed by the respective crystal structures of CbAR-H162F-NADP⁺-substrate complexes. Therefore, this study shows the great potential for engineering anthrol reductases to exploit their promiscuity, and then to prepare various complex compounds with multiple chiral centers with high enzyme activity and stereoselectivity.

Fig. 5 Molecular insights of asymmetric reduction of α -haloacetophenones for the preparation of β -halohydrins by CbAR-H162F. (a) Two possible stereoisomers generated by reduction of α -haloacetophenones. **1k–1o** represents 5 α -haloacetophenones used in this study. (b) Reduced compounds (**2k–2o**) obtained by the reduction of substrates (**1k–1o**) by CbAR-H162F. n.d. indicates not detected. (c) Substrate binding pocket of CbAR-H162F for substrate **1o** (PDB: 8HFK). Atoms S and Br are colored in yellow and green, respectively. The 2Fo – Fc electron density maps of substrate **1o** and Met202 was contoured at 1.0 σ in blue color. (d) Proposed proton shuttling mechanism and substrate binding mode of CbAR-H162F for the reduction of substrate **1o**. The proposed proton shuttling mechanism of CbAR towards substrate **1o** is indicated by the blue arrow.

Q22: Supplementary table 1- X-ray statistics:

I find it hard to believe that the B-factors for a 3.3 Angstrom structure are lower than those found in the 1.85 Angstrom structure and these B-factors don't agree with what is found in

the PDB reports. Speaking of the PDB reports, I find it remarkable that the emodin and NADP structures found in these reports are 'perfect'. There are no outliers or strained torsions, rings, bond angles and bond lengths- suggesting that these may not have been refined with the protein structures. This is also a little strange given that the structures were not refined with particularly tight constraints on the bond lengths or angles (see Supp Table 1 statistics regarding bond lengths and angles).

A22: Thank you very much for your wonderful suggestions. We have re-processed and re-uploaded two sets of data to ensure the ligands were refined together with the protein structures. In addition, we have checked and updated the X-ray statistics in Supplementary Table 2. New PDB validation reports are provided.

Q23: It is common to show difference density maps of substrates/ligands/cofactors found in the active site of enzymes and I suggest this would be a good addition to the manuscript so readers can see the density for themselves.

A23: Thank you very much again for your wonderful suggestions. As suggested, we have shown 2Fo – Fc electron density maps of emodin in Fig. 3b and Supplementary Fig. 3. Also, the electron density maps of other substrates (2,2-disubstituted 1,3-cyclodiketones and α -haloacetophenones) were provided according to your suggestions (Fig. 4, 5)

Q24: The authors have put together a nice manuscript describing their work and it is well written. I disagree with a few of the statements they've made regarding 'recruitment' of substrate into the active site and the importance of the hydrogen bond between His162 and Tyr210, but with further experiments they can prove my intuition incorrect. It would be good to show the density in the active site, particularly difference density for the emodin as well. It would also be good to have them check Table 1 X-ray statistics to make sure there are no typos and that they more closely align with the PDB reports.

A24: Thank you very much again for your positive evaluations and valuable suggestions. As suggested, we have changed the statement of "recruitment" to "recognition" in our revised manuscript and shown 2Fo – Fc electron density maps of emodin in Fig. 3b and Supplementary Fig. 3. In addition, we have checked and updated the X-ray statistics in Supplementary Table 2. New PDB validation reports are provided.

Reviewer 3: Hou et al reported the identification and characterization of a new anthrol reductase from fungi *Cercospora*, (CbAR). The enzyme has been shown to catalyze the reduction of substrates (emodin and estrones and analogues) using NADPH. The

reduction of the same substrates had also been shown in earlier studies where anthrol reductases such as MdpC, AfIM, Arti or ARti-2 were used.

To understand the mechanism of emodin hydroquinone by CbAR, they obtained the crystal structure co-crystallized with NADPH along with or without emodin, which gave vital information about the binding site and active site of the enzyme. The analysis of crystal structures of CbAR with and without emodin revealed the role of amino acid residues 209-216 in substrate recruitment. In addition, Gln152, His162 and Tyr210 have been shown to play crucial role in substrate recruitment and binding. This is a significant finding to understand the mechanism of functioning of this new class of enzymes, the anthrol reductases.

Authors also predict the possible reduction of 2,2-disubstituted 1,3-cyclodiketones by CbAR to obtain challenging 2,2-disubstituted-3-hydroxycycloketones as products. Accordingly, 2,2-disubstituted 1,3-cyclodiketones were reduced to obtained (2S, 3S)-ketols with high conversion and enantiomeric access. With the help of docking studies, they identified the key amino acid residues to further enhance the activity and there prepared mutants. The use of CbAR-H216F gave the corresponding (2S,3S)-ketodiols with >99% conversion. The CbAR-H162F mutant gave 44.7 fold increase compared to WT enzyme. These results suggests that further manipulation of 209-216 region may allow the reduction of other substrates by CbAR.

Overall it is a nice piece of work, where authors could give insight into the functioning of anthrol reductase CbAR for the first time. The binding of emodin at the active site gives crucial information about the reduction of anthrols at the active site. The reduction of 2,2-disubstituted 1,3-cyclodiketones to obtained (2S, 3S)-ketols is also a significant result. The work is of high significance to the field of biocatalysis. The data supports the findings of the work.

Based on the assessment and the work done I recommend the publication of the work in Nature Communications after revision as mentioned below.

A: Thank you for your nice and professional comments on our article. According to your suggestions, we have supplemented several new data here and corrected several mistakes in our previous draft. Based on your comments, here we prepared this point-to-point response and highlighted the answers in blue.

Q1: Page 1, line 1: Please mention the same title in manuscript and SI.

A1: Thank you very much for your careful correction! Due to the supplementary results of

the crystal structures and reduction reactions during the revision, we have changed both the title in manuscript and supporting information to “Structural analysis of an anthrol reductase inspires enantioselective synthesis of enantiopure hydroxycycloketones and β -halohydrins”, and make sure they are consistent.

Q2: line 16-19: Sentence “Then, crystal.....” is too long and confusing. It is better to rephrase it.

A2: Thank you very much for your suggestions. The long sentence has been rewritten as follows:

Then, crystal structures of CbAR and CbAR-Emodin complex were solved. It reveals that Tyr210 is critical for emodin recognition and binding, as it forms hydrogen-bond interaction with His162 and π - π stacking interaction with emodin, respectively. This ensures the correct orientation for the stereoselectivity.

Q3: Page 2, line 21: The conversion rates can be written as 93% instead of 92.9%.

A3: Thank you very much for your correction! As suggested, we have rounded all the conversion rate to integers in our manuscript and Fig. 1d, 4b, 5b, Supplementary Table 3 and Supplementary Table 5.

Q4: Page 3, line 37-40: The sentence “However, enzymes.....” is too long and confusing. Please replace it or divide into two sentences.

A4: Thank you very much for your suggestions. The long sentence has been rewritten as follows to avoid confusion:

However, enzymes used to synthesize optically pure 2,2-disubstituted-3-hydroxycycloketones from prochiral 2,2-disubstituted 1,3-cyclodiketones through enantioselective desymmetrization are still little reported.

Q5: Page 3, line 40: Use “this is because” instead use “The reason is that”

A5: Thank you very much for your suggestions. As suggested, “The reason is that” has been changed to “this is because” in revised manuscript and highlighted in red.

Q6: Page 4, line 70: please make it 93% instead of 92.9%. Apply same throughout the manuscript.

A6: Thank you very much for your correction. As suggested, we have rounded the

conversion rate to integers throughout the manuscript.

Q7: Page 4, line 75: “the SDR superfamily and shares the highest sequence identity of 86% with 17 β -HSDcl. Please also give sequence identity with other anthrol reductases. Can be include in SI.

A7: Thank you very much for wonderful suggestions. We have showed sequence identity of CbAR with selected 11 amino acid sequences in Supplementary Table 1.

Supplementary Table 1. Pairwise identity of CbAR with selected 11 related proteins.

Protein	Organism	Accession No.	Amino acid	Identity to CbAR (%)
17 β -HSDcl	Curvularia lunata	3IS3_A	270	86.6
ClaC	Passalora fulva	XP_047768478.1	267	84.2
ARti2	Talaromyces islandicus	CRG89873.1	265	83.9
CPUR 05429	Claviceps purpurea	M1W270.1	264	81.3
ARti	Talaromyces islandicus	CRG86682.1	265	79.2
MdpC	Aspergillus nidulans	XP_657750.1	265	74.6
AgnL6	Paecilomyces divaricatus	A0A411PQN6.1	268	68.9
T4HNR	Pyricularia oryzae	XP_003709023.1	283	63.3
T3HNR	Colletotrichum orbiculare	P87025.4	282	63.5
GDH	Bacillus megaterium	WP_013081865.1	261	30.2
CDH	Rhodococcus erythropolis	Q9RA05.1	277	27.4

Q8: Page 4, line 81: please modify “conversion rate” to “conversion”.

A8: Thank you very much for your suggestions. We have modified all the “conversion rate” to “conversion” throughout the manuscript.

Q9: Page 4, line 85: the temperature is written as 40.27 °C. Make it up to one decimal places

A9: Thank you very much for your suggestions. As suggested, we have changed 40.27 °C to 40.3 °C in our manuscript and Supplementary Fig. 1g and 1h.

Q10: Page 4, line 89: instead of “could” better to use “is known to catalyze”

A10: Thank you very much for your wonderful suggestions. As suggested, we have changed “could” to “is known to catalyze” in revised manuscript and highlighted in red.

Q11: Page 4, line 89-91: The sentence is not clear. Please rephrase the sentence or divide the content into two sentences.

A11: Thank you very much for your suggestions. The long sentence has been rewritten as follows:

We then determined whether estrone could be reduced by CbAR. As shown in Fig. 1d, 1b and its analogues (1c, 1d) could be correspondingly transformed to 17 β -estradiols with 99% de. In fact, they could be converted from related precursors 2,2-disubstituted-3-hydroxycycloketones.

Q12: Page 4, line 92-93: The inference drawn from the reduction of estrone and its analogues (1b, 1c, 1d) by CbAR to reduce 2,2-disubstituted 1,3-cyclodiketones is not very convincing. The authors have not look at the possibility of other related substrates. Although, 17b-HSD of *Curvularia lunata* has been shown to catalyze the reduction of estrones, its physiological function is more likely to be the reduction of anthrols.

Why authors have not considered six membered 1,3-diketones for reduction by CbAR. It is better to provide further justification for choosing 2,2-disubstituted 1,3-cyclodiketones as the substrates.

A12: Thank you very much for your wonderful suggestions. We regret for any confusion caused by our imprecise language. We have deleted the relevant content from the revised manuscript. In addition, according to your suggestion, we have tested a six membered 1,3-diketones (**1j**), both CbAR and CbAR-H162F showed high stereoselectivity (>99/1 dr, >99% ee) and low to moderate activity (up to 46% conversion) for this substrate (Fig. 4).

Fig. 4 Molecular insights of reductive desymmetrization of 2,2-disubstituted prochiral 1,3-cyclodiketones for the preparation of (2*S*, 3*S*)-2,2-disubstituted-3-hydroxycycloketones by CbAR-H162F. (a) Four possible stereoisomers generated by reduction desymmetrization of 2,2-disubstituted-1,3-cyclodiketones. **1e–1j** represent 6 prochiral 1,3-cyclodiketones used in this study. (b) Reduced compounds (**2e–2j**) obtained by the reduction of substrates (**1e–1j**) by CbAR-H162F. (c) Substrate binding pocket of CbAR-H162F for substrate **1e**. The 2*Fo* – *Fc* electron density maps of substrate **1e** was contoured at 1.0 σ in blue color (PDB: 8HFJ). (d) Proposed proton shuttling mechanism and substrate binding mode of CbAR-H162F for the reduction of substrate **1e**. The proposed proton shuttling mechanism of CbAR towards substrate **1e** is indicated by the blue arrow.

Q13: Page 8, line 172: “owing to two chiral centers of 2,2-disubstituted 1,3-cyclodiketones” need to be corrected.

A13: Thank you very much for your suggestions. We have corrected the description to “Owing to two chiral centers of 2,2-disubstituted-3-hydroxycycloketones, four stereoisomers could be possibly synthesized through reductive desymmetrization of 2,2-disubstituted 1,3-cyclodiketones (Fig. 4a)”.

Q14: Page 8, line 174-178. The sentence “Since, CbAR can accommodate……” is confusing and too long. please rephrase it.

A14: Thank you very much for your suggestions. The long sentence has been rewritten as follows:

Since CbAR can accommodate different bulky substrates (Fig. 1d), we next attempted to

investigate whether CbAR could convert 2,2-disubstituted 1,3-cyclodiketones into optically pure 2,2-disubstituted-3-hydroxycycloketones through reductive desymmetrization.

Q15: Page 8, line 182: “which could be used for the preparation of many natural products” how? If the authors can give the structures of the natural products such as such as cortistatin A, clavulactone, madindoline A, digitoxigenin and hygrophorone by highlighting the 2,2-disubstituted-3-hydroxycycloketones moiety, it will be helpful to understand the significance of the compounds that have been synthesized. This information may be included either in the manuscript or Supporting Information.

A15: Thank you very much for your wonderful suggestions. As suggested, we have given the structures mentioned above and highlighted the 2,2-disubstituted-3-hydroxycycloketones moiety in blue color in Supplementary Fig. 6.

Supplementary Figure 6. Representative natural products containing 2,2-disubstituted-3-hydroxycycloketones moiety. 2,2-disubstituted-3-hydroxycycloketones moiety is highlighted in blue color.

Q16: Page 9, line 200: replace 92.9% with 93%

A16: Thank you very much for your correction. As suggested, we have replaced 92.9% with 93%, and rounded the conversion rate to integers throughout the manuscript.

Q17: Page 14, line 335: The absolute configuration of 17b-esterdiol and its analogues was determined by comparison of previously reported ^1H NMR spectra. This statement is incorrect, please correct it.

A17: Thank you very much for your correction. In general, the absolute configuration can't be determined by comparing ^1H NMR spectra. However, estrone and its analogues bears several chiral centers, and the reduction of the carbonyl group will lead to the formation of two different diastereomers with different ^1H NMR spectra. Hence, we could determine the absolute configuration of 17β -estradiol and its analogues by comparison of previously

reported ^1H NMR spectra.

Therefore, to avoid the misunderstanding, we have changed the description to “The absolute configuration of 17β -estradiol and its analogues was determined by comparison with previously reported ^1H NMR data due to the diastereomers formed after the reduction of the carbonyl groups”.

Q18: S24-S37: The number and structures on the NMR spectra are not clearly visible. Please increase the font size and put structures in high resolution.

A18: Thank you very much for your suggestions. We have updated all NMR spectra with the increased font size and high resolution structures in Supporting Information.

REVIEWER COMMENTS

Reviewer #1 (Remarks to the Author):

The authors have addressed all of this reviewers comments which improves the manuscript significantly. The additional experiments also help to improve the novelty of the study now that additional data has been presented. Given the improvements it is possible the paper may be suitable for publication in Nature communications.

Reviewer #2 (Remarks to the Author):

Review of Structural analysis of an anthrol reductase to inspire an efficient reductive desymmetrization of bulky 1,3-cyclodiketones by Hou et al. – second version

Although the authors have made many commendable changes and added additional data to improve the manuscript, there are still a few issues (some newly added) that require attention prior to publication.

Firstly, the recommendation to show difference density maps, sometimes called Fo-Fc or mFo-DFc maps, has not been taken. The authors show 2Fo-Fc maps (or more specifically, 2mFo-DFc), which by their nature, are biased by the model. The difference maps should be done without the compound in the model and are traditionally contoured at 3 sigma to show the most significant features in the difference map. This is particularly important when investigators have partially ordered molecules such as that in figure 4c. Note: from the PDB validation file, only two of the possible four binding sites are occupied by the cyclodiketone and this should be noted in the text. These difference maps can be either put into the main text and exchanged with the current standard maps (figures 3b, 4c and 5c) or put in as additional figures in the supplementary material section.

Along with this recommendation, it would be good to show the fit of the mobile loop, residues Pro212-Pro219, that the authors discuss extensively, to the density in the emodin structure. The PDB validation file states that these residues don't fit the density well, so it would be good for readers to see how well ordered this loop is as significant discussion is based on the ordering of this loop in this structure. It could also be noted that the other compounds (cyclodiketone and the halogenated aryl ketone) seem to partially order this loop to different levels depending on the protomer (A,B,C, or D; when bound to one of the compounds) and what this may mean for the reaction, future engineering, etc.

There are some minor English changes that could be made, starting in the abstract:

The sentence (lines 16-18): "It reveals that Tyr210 is critical for emodin recognition and binding, as it forms hydrogen-bond interaction with His162 and π - π stacking interaction with emodin, respectively." Should probably be changed to "It reveals that Tyr210 is critical for emodin recognition and binding, as it forms a hydrogen-bond interaction with His162 and π - π stacking interactions with emodin."

End of line 57 should be: "these kinds of enzymes" or "this kind of enzyme" but not "this kind of enzymes".

Remove "respectively" from the end of the sentence in line 63.

Line 108: I feel it is necessary to point out that supplementary figure 2 does not show that CbAR folds as a tetramer. It only shows that there are four molecules in the crystallographic asymmetric unit. The number of protomers in the asymmetric unit generally has nothing to do with how a protein folds nor the folding process. If the authors wish to assert that CbAR folds as a tetramer, I believe they will need to do some additional experiments to prove this. This assertion also seems to be refuted by their experiments which show that the protein forms a dimer in solution...

Line 139: the start of the sentence should be "Asn152 is also involved in..." or "Asn152 is involved also in..."

Line 142: "...catalytic active residues Ser151 and Tyr165 with the distance of 2.6 Å and 2.5 Å, respectively." Should be "...catalytic active residues Ser151 and Tyr165 with distances of 2.6 Å and 2.5 Å, respectively."

Figure 4 and lines 217-218: the purported hydrogen bond between Tyr165 and the compound really can't be called that at 3.7 Angstroms, as that is about 0.5 Angstroms too long for this kind of bond.

From the PDB validation report, there are a few serious steric issues with the compound (L8U-halogenated aryl ketone) too close to Met202 with overlaps of 0.96 to >1.0 Angstroms. The bromine atom of the compound and the methionine sulfur atoms can't really be this close together. There are some serious steric clashes with the cyclodiketone compound as well (>1.0 Angstrom) that should be investigated. As mentioned in the previous review, their geometric constraints during refinement seem to be a little 'loose'. I appreciate that they have now included the ligands and cofactor in the refinement process. It can now be seen (PDB validation files) that these structures could be improved by having tighter geometric constraints (on both the compounds and on the protein). This is a subjective point and there are many opinions as to how tight restraints (or constraints) should be during the refinement process, but it is something to consider.

For Table 2 in the supplementary information- the X-ray statistics. As both tetragonal and orthorhombic space groups are defined by having all angles at 90 degrees, it isn't necessary to include the extra zeros (sometimes one, mostly two) in the table. 90, 90, 90 is fine and understood to be correct (and exact). The CC1/2 is an important number to include in the table. It is also traditional to include some additional numbers in this table that are specific to ligands/ water/ additional features in the model; more specifically, the number of compound atoms/cofactor atoms/waters and average B-factors of these compounds/cofactors/waters. This shows the reader how similar or different these B-factors are to those of the protein structure as a whole.

The authors have added substantial information with the two additional crystal structures and they have corrected many of the previous typos, grammatical errors and questionable statements. With a little additional work as described above, I believe they will have a substantially better manuscript for publication.

Reviewer #3 (Remarks to the Author):

The authors have satisfactorily addressed the comments and suggestions, which has improved the manuscript considerably. I recommend it for publication in Nature Communications in the current form.

Response to reviewers:

To Reviewer 1:

The authors have addressed all of this reviewers comments which improves the manuscript significantly. The additional experiments also help to improve the novelty of the study now that additional data has been presented. Given the improvements it is possible the paper may be suitable for publication in Nature communications.

A: We appreciate your positive comment and are pleased that we have addressed all points to your satisfaction.

To Reviewer 2:

Review of Structural analysis of an anthrol reductase to inspire an efficient reductive desymmetrization of bulky 1,3-cyclodiketones by Hou et al. – second version

Although the authors have made many commendable changes and added additional data to improve the manuscript, there are still a few issues (some newly added) that require attention prior to publication.

A: We appreciate your positive comments and valuable suggestions again, which help us further improve the quality of this manuscript. Here we prepared the point-to-point response as you suggested.

Q1: Firstly, the recommendation to show difference density maps, sometimes called Fo-Fc or mFo-DFc maps, has not been taken. The authors show 2Fo-Fc maps (or more specifically, 2mFo-DFc), which by their nature, are biased by the model. The difference maps should be done without the compound in the model and are traditionally contoured at 3 sigma to show the most significant features in the difference map. This is particularly important when investigators have partially ordered molecules such as that in figure 4c. Note: from the PDB validation file, only two of the possible four binding sites are occupied by the cyclodiketone and this should be noted in the text. These difference maps can be either put into the main text and exchanged with the current standard maps (figures 3b, 4c and 5c) or put in as additional figures in the supplementary material section.

A1: Thank you very much for your professional advice again. We apologize for the confusion between Fo-Fc and 2Fo-Fc maps. As suggested, the Fo – Fc omit maps of substrates, which were contoured at 3.0 σ , has been shown in Supplementary Fig. 3.

Additionally, the polder maps (an improved omit map) contoured at 3.0σ were also provided in Supplementary Fig. 3. Together with the 2Fo-Fc maps, now we can clearly show that all substrates could well localize at the catalytic sites and explain the catalytic mechanisms.

Supplementary Figure 3. The difference density maps of substrates in different crystal structures. (a) The Fo – Fc omit map of emodin was contoured at 3.0σ in green color. (b) The polder map of emodin was contoured at 3.0σ in green color. (c) The Fo – Fc omit map of substrate **1e** was contoured at 3.0σ in green color. (d) The polder map of substrate **1e** was contoured at 3.0σ in green color. (e) The Fo – Fc omit map of substrate **1o** was contoured at 3.0σ in green color. (f) The polder map of substrate **1o** was contoured at 3.0σ in green color. NADP⁺, substrates, the catalytic triad (Ser151, Tyr165 and Lys169), and residues involved in binding of substrates are highlighted in different colors.

In addition, the sentence “only two of the possible four binding sites are occupied by **1e**”

was accordingly updated in the manuscript as follow:

the 2-methyl-cyclopentane-1,3-dione part of **1e** is clearly observed and well localized to the active site of CbAR-H162F (Fig. 4c, d and Supplementary Fig. 3c, d). Notably, we observed that only two of the possible four binding sites in the crystallographic asymmetric unit are occupied by **1e**.

Q2: Along with this recommendation, it would be good to show the fit of the mobile loop, residues Pro212-Pro219, that the authors discuss extensively, to the density in the emodin structure. The PDB validation file states that these residues don't fit the density well, so it would be good for readers to see how well ordered this loop is as significant discussion is based on the ordering of this loop in this structure. It could also be noted that the other compounds (cyclodiketone and the halogenated aryl ketone) seem to partially order this loop to different levels depending on the protomer (A, B, C, or D; when bound to one of the compounds) and what this may mean for the reaction, future engineering, etc.

A2: Thank you very much for your great suggestions! As suggested, the residues and electron density around the flexible region (between His209 and Pro219) in the different crystal forms have been provided in Supplementary Fig. 4. We believe that these informations will help the readers observe this region clearly. Indeed, as you observed, upon the binding of different substrates, the conformation of this region is different, suggesting the future engineering of this kind of enzyme. Thank you again for your valuable suggestions, this flexible region will be studied in-depth in our future work.

Supplementary Figure 4. The residues and electron density around the flexible region

(between His209 and Pro219) in the different crystal forms. (a) The residues and 2Fo – Fc electron density map contoured at 1.0 σ around the flexible region in the structure of CbAR-NADP⁺-Emodin complex are shown. (b) The residues and 2Fo – Fc electron density map contoured at 1.0 σ around the flexible region in the structure of CbAR-H162F-NADP⁺-1e complex are shown. (c) The residues and 2Fo – Fc electron density map contoured at 1.0 σ around the flexible region in the structure of CbAR-H162F-NADP⁺-1o complex are shown.

Q3: There are some minor English changes that could be made, starting in the abstract: The sentence (lines 16-18): “It reveals that Tyr210 is critical for emodin recognition and binding, as it forms hydrogen-bond interaction with His162 and π - π stacking interaction with emodin, respectively.” Should probably be changed to “It reveals that Tyr210 is critical for emodin recognition and binding, as it forms a hydrogen-bond interaction with His162 and π - π stacking interactions with emodin.”

A3: Thank you very much for your correction. We have corrected it as suggested.

Q4: End of line 57 should be: “these kinds of enzymes” or “this kind of enzyme” but not “this kind of enzymes”.

A4: Thank you very much again for your correction. As suggested, we have changed to “this kind of enzyme”

Q5: Remove “respectively” from the end of the sentence in line 63.

A5: Thank you very much for your suggestions. We have removed “respectively” from the end of the sentence in line 63.

Q6: Line 108: I feel it is necessary to point out that supplementary figure 2 does not show that CbAR folds as a tetramer. It only shows that there are four molecules in the crystallographic asymmetric unit. The number of protomers in the asymmetric unit generally has nothing to do with how a protein folds nor the folding process. If the authors wish to assert that CbAR folds as a tetramer, I believe they will need to do some additional experiments to prove this. This assertion also seems to be refuted by their experiments which show that the protein forms a dimer in solution.

A6: Thank you very much again for your correction. We apologize for the confusion of tetramer. According to your comments, we have rewritten this sentence as follow.

“It shows that the crystallographic asymmetric unit contains four molecules (Supplementary Fig. 2), but CbAR was eluted as a dimer in solution (Supplementary Fig. 1b, c).”

Q7: Line 139: the start of the sentence should be “Asn152 is also involved in...” or “Asn152 is involved also in...”

A7: Thank you very much again for your wonderful suggestions. As suggested, we have rewritten this sentence as “Asn152 is also involved in the binding of emodin through the hydrogen bond interaction.”

Q8: Line 142: “...catalytic active residues Ser151 and Tyr165 with the distance of 2.6 Å and 2.5 Å, respectively.” Should be “...catalytic active residues Ser151 and Tyr165 with distances of 2.6 Å and 2.5 Å, respectively.”

A8: Thank you very much for your suggestions. We have corrected it as suggested

Q9: Figure 4 and lines 217-218: the purported hydrogen bond between Tyr165 and the compound really can't be called that at 3.7 Angstroms, as that is about 0.5 Angstroms too long for this kind of bond.

A9: Thank you very much for your correction! According to your suggestion, we have rewritten this sentence as “One carbonyl groups of **1e** forms the hydrogen-bond interactions with the amino group of Gly197 (Fig. 4c, d)”, and updated the fig. 4c as follow.

Fig. 4 (c) Substrate binding pocket of CbAR-H162F for substrate **1e**. The $2F_o - F_c$ electron density maps of substrate **1e** was contoured at 1.0σ in blue color (PDB: 8HFJ).

Q10: From the PDB validation report, there are a few serious steric issues with the compound (L8U- halogenated aryl ketone) too close to Met202 with overlaps of 0.96 to >1.0 Angstroms. The bromine atom of the compound and the methionine sulfur atoms can't really be this close together. There are some serious steric clashes with the cyclodiketone compound as well (>1.0 Angstrom) that should be investigated. As

mentioned in the previous review, their geometric constraints during refinement seem to be a little 'loose'. I appreciate that they have now included the ligands and cofactor in the refinement process. It can now be seen (PDB validation files) that these structures could be improved by having tighter geometric constraints (on both the compounds and on the protein). This is a subjective point and there are many opinions as to how tight restraints (or constraints) should be during the refinement process, but it is something to consider.

A10: Thank you very much for your professional correction again! We truly learn a lot from your professional comments. According to your new suggestions, we have re-investigated the steric issues according to your suggestion. New fig. 4c and fig. 5c in the suitable positions are provided. In addition, we have checked and updated the X-ray statistics in Supplementary Table 2. New PDB validation reports are also provided.

Fig. 4 (c) Substrate binding pocket of CbAR-H162F for substrate **1e**. The $2F_o - F_c$ electron density map of substrate **1e** was contoured at 1.0σ in blue color (PDB: 8HFJ).

Fig. 5 (c) Substrate binding pocket of CbAR-H162F for substrate **1o** (PDB: 8HFK). Atoms

S and Br are colored in yellow and green, respectively. The 2Fo – Fc electron density map of substrate **1o** and Met202 was contoured at 1.0 σ in blue color.

Supplementary Table 2. X-ray data collection and refinement statistics.

	CbAR-NADP ⁺	CbAR-NADP ⁺ - Emodin	CbAR-H162F- NADP ⁺ - 1e	CbAR-H162F- NADP ⁺ - 1o
PDB code	7YB1	7YB2	8HFJ	8HFK
Data collection				
Space group	P 4 ₁ 2 ₁ 2	P 2 2 ₁ 2 ₁	P 4 ₁ 2 ₁ 2	P 4 ₁ 2 ₁ 2
Cell dimensions				
a , b , c (Å)	124.62, 124.62, 134.21	66.80, 123.36, 126.11	124.92, 124.92, 133.78	124.87, 124.87, 133.19
α , β , γ (°)	90, 90, 90	90, 90, 90	90, 90, 90	90, 90, 90
Resolution (Å)	23.66 - 3.30 (3.42 - 3.30)	45.89 - 1.85 (1.92 - 1.85)	24.06 – 2.75 (2.848 – 2.75)	24.04 – 2.90 (3.00 – 2.90)
R _{merge}	0.29 (0.48)	0.08 (0.61)	0.20 (0.58)	0.23 (0.77)
I / σ (I)	7.2 (4.3)	32.6 (3.3)	7.9 (2.7)	8.3 (2.6)
CC _{1/2}	0.912 (0.819)	0.996 (0.868)	0.983 (0.827)	0.987 (0.827)
Completeness (%)	99.5 (99.9)	98.74 (89.96)	99.69 (99.86)	99.67 (99.96)
Redundancy	10.4 (8.5)	12.1 (7.1)	6.8 (5.3)	9.8 (8.6)
Refinement				
No. reflections	16410 (1595)	88466 (7927)	28077 (2777)	23869 (2325)
R _{work} /R _{free}	0.206/0.285	0.151/0.188	0.205/0.270	0.202/0.289
No. atoms				
Protein	7525	7918	7669	7595
Ligand	192	296	224	240
Water	17	569	103	55
B-factors (Å²)				
Protein	17.4	21.2	21.9	30.8
Ligand	13.4	20.8	22.0	31.8
Water	26.7	29.3	21.9	22.9
R.m.s. deviations				
Bond lengths (Å)	0.015	0.015	0.015	0.015
Bond angles (°)	1.94	1.85	1.93	1.87
Ramachandran outliers (%)	0.51	0.00	0.00	0.40
Ramachandran favored (%)	90.67	96.72	94.55	91.72

The values in parentheses are for highest-resolution shell.

Q11: For Table 2 in the supplementary information- the X-ray statistics. As both tetragonal

and orthorhombic space groups are defined by having all angles at 90 degrees, it isn't necessary to include the extra zeros (sometimes one, mostly two) in the table. 90, 90, 90 is fine and understood to be correct (and exact). The $CC_{1/2}$ is an important number to include in the table. It is also traditional to include some additional numbers in this table that are specific to ligands/ water/ additional features in the model; more specifically, the number of compound atoms/cofactor atoms/waters and average B -factors of these compounds/cofactors/waters. This shows the reader how similar or different these B -factors are to those of the protein structure as a whole.

A11: Thank you very much again for your professional suggestions. We truly learn a lot from your professional comments.

(1) As suggested, we have changed 90.00 and 90.0 degrees to 90 in Supplementary Table 2.

(2) $CC_{1/2}$, the number of compound atoms/cofactor atoms/waters and average B -factors of these compounds/cofactors/waters were also included in the updated Supplementary Table 2 and highlighted in red. Notably, although we have re-processed the data, the average B -factors of compounds/cofactors/waters for CbAR-NADP⁺ (3.3 Å resolution) are still lower than that of CbAR-NADP⁺-Emodin (1.85 Å resolution). The possible reason might come from the flexible region (P209-T216), which was omitted in CbAR-NADP⁺ complex, more waters and the substrate emodin in CbAR-NADP⁺-Emodin complex (Supplementary Table 2 and Fig. R1),

Figure R1. B -factors of the flexible region (P209-T216) in CbAR-NADP⁺-Emodin.

Supplementary Table 2. X-ray data collection and refinement statistics.

	CbAR-NADP ⁺	CbAR-NADP ⁺ - Emodin	CbAR-H162F- NADP ⁺ -1e	CbAR-H162F- NADP ⁺ -1o
PDB code	7YB1	7YB2	8HFJ	8HFK
Data collection				
Space group	P 4 ₁ 2 ₁ 2	P 2 2 ₁ 2 ₁	P 4 ₁ 2 ₁ 2	P 4 ₁ 2 ₁ 2
Cell dimensions				
a , b , c (Å)	124.62, 124.62, 134.21	66.80, 123.36, 126.11	124.92, 124.92, 133.78	124.87, 124.87, 133.19
α , β , γ (°)	90, 90, 90	90, 90, 90	90, 90, 90	90, 90, 90
Resolution (Å)	23.66 - 3.30 (3.42 - 3.30)	45.89 - 1.85 (1.92 - 1.85)	24.06 – 2.75 (2.848 – 2.75)	24.04 – 2.90 (3.00 – 2.90)
R _{merge}	0.29 (0.48)	0.08 (0.61)	0.20 (0.58)	0.23 (0.77)
I / σ (I)	7.2 (4.3)	32.6 (3.3)	7.9 (2.7)	8.3 (2.6)
CC _{1/2}	0.912 (0.819)	0.996 (0.868)	0.983 (0.827)	0.987 (0.827)
Completeness (%)	99.5 (99.9)	98.74 (89.96)	99.69 (99.86)	99.67 (99.96)
Redundancy	10.4 (8.5)	12.1 (7.1)	6.8 (5.3)	9.8 (8.6)
Refinement				
No. reflections	16410 (1595)	88466 (7927)	28077 (2777)	23869 (2325)
R _{work} /R _{free}	0.206/0.285	0.151/0.188	0.205/0.270	0.202/0.289
No. atoms				
Protein	7525	7918	7669	7595
Ligand	192	296	224	240
Water	17	569	103	55
B-factors (Å²)				
Protein	17.4	21.2	21.9	30.8
Ligand	13.4	20.8	22.0	31.8
Water	26.7	29.3	21.9	22.9
R.m.s. deviations				
Bond lengths (Å)	0.015	0.015	0.015	0.015
Bond angles (°)	1.94	1.85	1.93	1.87
Ramachandran outliers (%)	0.51	0.00	0.00	0.40
Ramachandran favored (%)	90.67	96.72	94.55	91.72

The values in parentheses are for highest-resolution shell.

Q12: The authors have added substantial information with the two additional crystal structures and they have corrected many of the previous typos, grammatical errors and questionable statements. With a little additional work as described above, I believe they will have a substantially better manuscript for publication.

A12: Thank you very much again for your positive comments and careful review. As suggested, we have shown Fo – Fc omit maps and polder maps of emodin, substrate **1e** and substrate **1o** in Supplementary Fig. 3. We have also shown the residues and electron density around the flexible region (between His209 and Pro219) in the different crystal forms in Supplementary Fig. 4. In addition, we have addressed the steric issues towards two substrates **1e** and **1o**, and updated the X-ray statistics in Supplementary Table 2.

To Reviewer 3:

The authors have satisfactorily addressed the comments and suggestions, which has improved the manuscript considerably. I recommend it for publication in Nature Communications in the current form.

A: Thank you for your positive assessment for our revision, and we are pleased that we were able to address all points to your satisfaction.

REVIEWERS' COMMENTS

Reviewer #2 (Remarks to the Author):

The authors have modified the manuscript to include the suggested alterations and updates. I appreciate the extra work this involved but I believe it makes for a more informative (and readable) paper.

I can now recommend publication.